**Comparison of hourly surface downwelling solar radiation estimated from MSG/SEVIRI and**
**forecast by RAMS model with pyranometers over Italy**
Stefano Federico[1], Rosa Claudia Torcasio[2], Paolo Sanò[1], Daniele Casella[1], Monica Campanelli[1], Jan
Fokke Meirink[3], PingWang[3], Stefania Vergari[4], Henri Diémoz[5], Stefano Dietrich[1]
(1) ISAC-CNR, via del Fosso del Cavaliere 100, 00133 Rome, Italy
(2) ISAC-CNR, zona Industriale comparto 15, 88046 Lamezia Terme, Italy
(3) Royal Netherlands Meteorological Institute (KNMI), Utrechtseweg 297, De Bilt, The
Netherlands
(4) Technical Centre for Meteorology (CTM), ITAF, Italy
(5) ARPA Valle D'Aosta, Italy
**ABSTRACT**
In this paper, we evaluate the performance of two Global Horizontal solar Irradiance (GHI)
estimates, one derived from Meteosat Second Generation (MSG) and another from the one-day
forecast of the Regional Atmospheric Modeling System (RAMS) mesoscale model. The horizontal
resolution of the MSG-GHI is $3*5$ km$^2$ over Italy, which is the focus area of this study. For this
paper, RAMS has the horizontal resolution of 4km.
The performance of MSG-GHI estimate and RAMS-GHI one-day forecast are evaluated for one
year (1 June 2013 – 31 May 2014) against data of twelve ground based pyranometers over Italy
spanning a range of climatic conditions, i.e. from maritime Mediterranean to Alpine climate.
Statistics for hourly GHI and daily integrated GHI are presented for the four seasons and the whole
year for all the measurement sites. Different sky conditions are considered in the analysis.
Results for hourly data show an evident dependence on the sky conditions, with the Root Mean
Square Error (RMSE) increasing from clear to cloudy conditions. The RMSE is substantially higher
for Alpine stations in all the seasons, mainly because of the increase of the cloud coverage for these
stations, which is not well represented at the satellite and model resolutions.
Considering the yearly statistics computed from hourly data for the RAMS model, the RMSE
ranges from 152 W/m$^2$ (31%) obtained for Cozzo Spadaro, a maritime station, to 287 W/m$^2$ (82%)
for Aosta, an Alpine site. Considering the yearly statistics computed from hourly data for MSG-
GHI, the minimum RMSE is for Cozzo Spadaro (71 W/m$^2$, 14%), while the maximum is for Aosta
(181 W/m$^2$, 51%). The Mean Bias Error (MBE) shows the tendency of RAMS to over forecast the
GHI, while no specific behaviour if found for MSG-GHI.

Results for daily integrated GHI show lower RMSE compared to hourly GHI evaluation for both RAMS-GHI one-day forecast and MSG-GHI estimate. Considering the yearly evaluation, the RMSE of daily integrated GHI is at least 9% lower (in percentage units, from 31% to 22% for RAMS in Cozzo Spadaro) than the RMSE computed for hourly data for each station. A partial compensation of underestimation and overestimation of the GHI contributes to the RMSE reduction. Furthermore, a post-processing technique, namely Model Output Statistics (MOS), is applied to improve the GHI forecast at hourly and daily temporal scales. The application of MOS shows an improvement of RAMS-GHI forecast, which depends on the site considered, while the impact of MOS on MSG-GHI RMSE is small.

## 1. Introduction

The Global Horizontal Irradiance (GHI) is the power of the solar spectrum reaching the surface and it is a key parameter for several disciplines. In particular, the exploitation of solar energy, which is the most abundant renewable energy, is of great interest because the larger penetration of renewable energies into the energy market would reduce the emissions of greenhouse gases (Szuromi et al 2007; IEA, 2010; EWEA, 2011) caused by human activities.

Photovoltaic (PV) systems enable the conversion of the solar radiation into electricity through semi-conductor devices and, in order to control the increase of global temperature, PV systems are expected to have a potential by more than 200 GW by 2020 (EWEA, 2011).

For the operation and implementation of PV systems, observation and forecast of GHI play a major role. Surface weather stations equipped with a pyranometer give reliable observations of GHI, but they are often unavailable in the places where new installations are planned. For this purpose, the GHI may be derived from other sources, as the Meteosat Second Generation (MSG) Spinning Enhanced Visible and Infrared Imager (SEVIRI) or a Numerical Weather Prediction Model (NWP).

In this paper, we show the performance of both the MSG-GHI estimate, following the methodology of Greuell et al. (2013), and RAMS-GHI one-day forecast over the whole Italian territory. To verify GHI, we use twelve pyranometers, which are representative of sites with very different climates, from Mediterranean maritime to Alpine. Moreover, the study spans a whole year to properly account for the natural variability of the Mediterranean climate.

Many studies are available on the performance of different approaches to estimate and forecast solar radiation in several countries in Europe (Roebeling et al, 2008; Greuell et al, 2013; Lara-Fanego et al., 2012; Kosmopulos et al., 2015; Gómez et al., 2016; Lorenz et al, 2009; Perez et al, 2006; Rincon et al, 2011), because the planning of new PV systems and the managing of the electricity

grid with large amounts of production from solar energy requires the knowledge and forecast of GHI with high accuracy. This study goes in this direction by considering a nation-wide evaluation for a whole year. Moreover, Italy has a great potential for the exploitation of solar energy (Petrarca et al., 2000).

We consider both the hourly and daily integrated GHI, the latter being the GHI integrated for each day for the different datasets, to evaluate the performance of both RAMS-GHI and MSG-GHI for two different timescales of interest. Also, we show the impact of a simple post processing technique, which aims to reduce the Mean Bias Error (MBE) for each site, on the GHI estimate and forecast.

The paper is organized as follows: Section 2 shows the datasets used and the methodology adopted to evaluate the errors of the MSG-GHI estimate and RAMS-GHI one-day forecast; Section 3 shows the results considering both the hourly and daily integrated GHI; Conclusions are given in Section 4.

## 2. Data and methods

### 2.1 Cloud properties and GHI from MSG-SEVIRI

The SEVIRI instrument onboard MSG carries 11 channels in the visible to infrared spectral range with a spatial resolution of 3x3 km$^2$ at the sub-satellite point and a temporal repeat frequency of 15 minutes. Over Italy the spatial resolution is about 3x5 km$^2$. From the SEVIRI measurements, a range of cloud physical properties can be derived with the Cloud Physical Properties (CPP) algorithm. The algorithm first identifies cloudy and cloud contaminated pixels using a series of thresholds and spatial coherence tests on the measured visible and infrared radiances (Roebeling et al., 2008). Depending on the tests, the sky can be classified as clear, contaminated or overcast. Subsequently, cloud optical properties (optical thickness) are retrieved by matching observed reflectances at visible (0.6 μm) and near-infrared (1.6 μm) wavelengths to simulated reflectances of homogeneous clouds composed of either liquid or ice particles. A mixture of ice and water is not possible in this framework. The thermodynamic phase (liquid or ice) is determined as part of this procedure, using a cloud-top temperature estimate as additional input (Roebeling et al., 2008; Stengel et al., 2014).

Building on the retrieval of cloud physical properties, the Surface Insolation under Clear and Cloudy Skies (SICCS) was developed to estimate surface downwelling solar radiation using broad-

band radiative transfer simulations (Deneke et al., 2008; Greuell et al., 2013). Both global irradiance as well as the direct and diffuse components are retrieved. While the cloud properties are the main input for cloudy and cloud-contaminated pixels, information about atmospheric aerosol from the Monitoring Atmospheric Composition and Climate (MACC) project is used for cloud-free scenes. The retrieval of cloud properties can be associated with large uncertainties, in particular due to horizontal inhomogeneity (e.g., Coakley et al., 2005). However, subsequently derived irradiances (such as SICCS GHI) have relatively much smaller uncertainty due to compensation of errors in forward and inverse radiative transfer calculations (Greuell et al., 2013; see also Kato et al., 2006). Uncertainties in MACC reanalysed aerosol properties contribute to errors in retrieved clear-sky GHI but these errors are considerably smaller than those for cloudy skies (Greuell et al., 2013).

Greuell et al. (2013) performed an extensive validation of the MSG-SICCS retrievals with Baseline Surface Radiation Network (BSRN) ground-based observations in Europe for the year 2006. They found median values of the station GHI biases of +7 W/m$^2$ (+2%) and hourly GHI RMSEs of 65 W/m$^2$ (18%).

The CPP and SICCS products are publicly available at msgcpp.knmi.nl.

*2.2 The RAMS set-up*

In this paper, we evaluate the performance of the RAMS-GHI one-day forecast. RAMS is a general purpose limited area model designed to be used at the mesoscale (horizontal grid spacing ≈ 1-100 km) or higher horizontal resolutions. It is based on a full set of non-hydrostatic, compressible equations of the atmospheric dynamics and thermodynamics, plus conservation equations for scalar quantities such as water vapour and liquid and ice hydrometeor mixing ratios. The model is widely used for research as well as for weather forecast (Cotton et al., 2003).

The model is run with two one-way nested grids (Table 1, Figure 1). The coarser domain has 12 km horizontal resolution and covers Central Europe, while the second domain has 4 km horizontal resolution and covers the Italian peninsula. Thirty-six vertical levels, extending up to the lower stratosphere, are used in the terrain-following coordinate system of RAMS.

The exchange between the atmosphere, the surface and the soil is computed by the LEAF (Land Ecosystem-Atmosphere Feedback) submodel. The LEAF submodel considers the interaction among several features, as well as their influence on the atmosphere: vegetation, soil, lakes and oceans, and snow cover.

RAMS parameterises the unresolved transport using $K$-theory, in which the covariance is evaluated
as the product of an eddy mixing coefficient and the gradient of the transported quantity. The
turbulent mixing in the horizontal directions relates the mixing coefficients to the fluid strain rate
(Smagorinsky, 1963) and includes corrections for the influence of the Brunt-Vaisala frequency and
the Richardson number (Pielke, 2002).
Convective precipitation is parameterised following Molinari and Corsetti (1985), who modified the
Kuo scheme (Kuo, 1974) to account for downdrafts. The convective scheme is applied to the
coarser RAMS domain, while convection is assumed explicitly resolved for the inner domain.
Explicitly resolved precipitation is computed by the WRF (Weather Research and Forecasting
System) – single-moment-microphysics class 6 (WSM6) scheme (Hong et al., 2006), which was
recently adapted to RAMS (Federico, 2016).
Short wave and long wave radiation is computed by the Chen and Cotton scheme (Chen and Cotton,
1983); the radiative scheme accounts for the total condensate in the atmosphere but not for the
specific hydrometeor type. In particular, the scheme uses an "effective emissivity" for cloud layers,
where the cloud emissivity is parametrized empirically from observations (Stephens 1978). The
"effective emissivity" is a function of the total condensate water path, computed summing all
hydrometeors mixing ratios for each model level (liquid, i.e. cloud and rain, solid, i.e. ice and snow,
and mixed phase, i. e. graupel) and integrating over the cloud-layer (Chen and Cotton, 1983).
Initial and boundary conditions are interpolated from the operational analysis/forecast cycle issued
at 12:00 UTC by the ECMWF (European Centre for Medium range Weather Forecast). Initial and
boundary conditions are available at 0.5° horizontal resolution and on nine pressure levels, from
1000 to 30 hPa. No additional data are assimilated into the RAMS model.
The model was run for a whole year (1 June 2013 - 31 May 2014) with the above configuration and
with no hydrometeors at the initial time, with the exception of water vapour (cold start). Previous
unpublished studies with RAMS showed that 12 h are enough for the model to reach a dynamical
equilibrium between the dynamic, thermodynamic and cloud-precipitation fields starting from a
cold start. For this reason, each simulation lasts 36 h, starts at 12 UTC of the day before the day of
interest, and the first 12 h are used as spin-up time and discarded. The model output is available
hourly.

*2.3 Surface observations*

In this work, we consider 12 pyranometers over Italy (Figure 2). Their coordinates, height above the sea level, the Institution responsible for their management, and abbreviations used in this paper are shown in Table 2. The pyranometers span a wide range of climatic conditions: Trapani, Cozzo Spadaro, Santa Maria di Leuca, Capo Palinuro, Pratica di Mare, Cervia, Pisa and Trieste are located by the sea, and show a typical Mediterranean climate; Vigna di Valle is characterized by a mild Mediterranean climate but it is located in more complex hilly terrain; Paganella, Monte Cimone and Aosta are mountainous stations, and this has an important impact on the RAMS and MSG performance at the sites. More specifically, Paganella is on the Alps, Monte Cimone is on the Apennines, while Aosta, while at lower altitude, is embedded in the rough Alpine terrain.

The pyranometers are managed by two different institutions. The Aosta pyranometer is managed by Arpa Valle D'Aosta, while all other pyranometers are managed by the Italian Air Force (Aeronautica Militare). Each institution is responsible for its own measurements.

For pyranometers managed by the Italian Air Force, in addition to basic maintenance and installing procedures recommended by WMO – Guide nr. 8, data quality is controlled following an internal control procedure described in Vergari et al. (2010).

In particular, to improve quality control checks for global solar radiation and sunshine duration data (available simultaneously for all stations of this paper managed by Aeronautica Militare), two procedures have been implemented. A range limit check, applied to both variables separately, concerns the respect of variables' physical limits. This check has been improved varying physical limits in agreement to the latitude and the season. Furthermore, the monthly atmospheric clearness index has been calculated from the climatic history of each site, by applying the linear form of the Angstrom-Prescott model. Then, an upper and a lower bound for the solar radiation are defined as linear functions of clearness index and the sunshine duration value. These bounds delimit the range of the daily solar radiation.

Analyzing the distance of daily values from their bounds, it is also possible to prevent instrumental electronic drifts. In fact, if this distance changes in an appreciable way, a recalibration procedure is activated and the device is recalibrated by comparison with a standard pyranometer using the sun as a source, under natural conditions of exposure (ISO ,1993). The reference standard used in this case is a CM11 Kipp and Zonen, calibrated every two year by the WMO Regional Instrument Centre Radiation of Carprentrass (France), by comparison with a pyreliometer PMO6 and a pyranometer CMP21.

For the Aosta pyranometer, in addition to the manual maintenance related to the periodical cleaning of the dome, irradiance measurements are daily checked through comparison with clear-sky

simulations by a radiative transfer model (libRadtran, Emde et al., 2016) to check for electric wiring faults. In particular, measurements higher than 200% of the daily maximum expected from libRadtran in clear-sky conditions are removed. The CMP21 radiometer is calibrated every two years at the Physikalisch-Meteorologisches Observatorium Davos/World Radiation Center (PMOD/WRC) against a member of the World Standard Group (WSG) for the direct component and a shaded standard pyranometer of the World Radiation Center (WRC) for the diffuse component. The radiometric stability was better than 0.2% over the period of the six years of measurements.

Table 3 shows, for each station and season, as well as for the whole year, the percentage of data in clear, contaminated and overcast conditions, classified by the satellite method of Section 2.1.

There is a considerable variability of the sky conditions with the season for each station. For Trapani, for example, the percentage of clear sky in summer is 82%, while it reduces to 38% in fall and 48% in winter. Also, for each season, the variability of the sky conditions among the stations is high. For maritime stations, for example, the percentage of clear skies in summer is above 70% with few exceptions, while it reduces to 45, 34, 32% for Paganella, Monte Cimone and Aosta, respectively.

*2.4 Evaluation methodology*

The RAMS GHI forecast is available hourly, while the frequency of pyranometer observations and MSG-GHI estimate is every half an hour. Pyranometer observations and MSG-GHI estimates were considered hourly, at the same time of the RAMS forecast output. Starting from these data, the MBE (Mean Bias Error) and the RMSE (Root Mean Square Error) were computed:

$$MBE = \frac{1}{N}\sum_{i=1}^{N}(x_{fi} - x_{oi}) \tag{1}$$

$$RMSE = \sqrt{\frac{1}{N}\sum_{i=1}^{N}(x_{fi} - x_{oi})^2} \tag{2}$$

Where $x_f$ is the RAMS forecast or the MSG GHI estimate, $x_o$ is the pyranometer observation, and $N$ is the total number of data available for the statistic.

In addition to the MBE and RMSE computed from hourly data, the statistics are computed starting from daily data. In this case, the integral of the GHI for the whole day is first computed for each dataset, then the MBE and RMSE are computed from the daily data.

Relative MBE and relative RMSE error measures (rMBE, rRMSE) are also used. The normalization is done with the pyranometer observation for the station and period considered, i.e. :


$$rMBE = 100\frac{\sum_{i=1}^{N}(x_{fi}-x_{oi})}{\sum_{i=1}^{N}x_{oi}} \tag{3}$$


$$rRMSE = 100\frac{\sqrt{\frac{1}{N}\sum_{i=1}^{N}(x_{fi}-x_{oi})^2}}{\frac{1}{N}\sum_{i=1}^{N}x_{oi}} \tag{4}$$


In order to improve the RAMS one-day hourly forecast and the MSG-GHI estimate, a post-
processing technique, namely the Model Output Statistics (MOS), is used. The MOS technique
improves the forecast/estimate of the GHI by reducing the MBE. The MBE is caused by several
factors related to both modelling and observations. In the context of this paper the most important
causes of MBE are: a) the approximations in the meteorological model and in the methodology used
to estimate GHI from MSG data, and; b) the horizontal grid used to represent the real world, which
smoothens the surface features causing systematic errors. Other contributions arise from small and
undetected systematic errors in the observations, and from the not exact simultaneity of the three
datasets (pyranometers, MSG-GHI, RAMS-GHI forecast).
The MOS used here consists of a linear regression computed between the GHI forecast (or estimate)
and observation for a training period:
$$y=a+bx \tag{5}$$
where $x$ is the RAMS-GHI one-day hourly forecast (or MSG hourly estimate) and $y$ is the
pyranometer observation. The application of the MOS is described in Section 3.4.

**3. Results**
**3.1 General considerations on MSG estimate and RAMS forecast**
Figure 3a shows the scatter-plot of hourly GHI estimates of MSG and the corresponding
pyranometer observations for Vigna di Valle. The black dots refer to clear sky, while the red dots
are for contaminated and overcast conditions (after also referred to as cloudy conditions) for the
entire yearly dataset. Three regression curves are shown: the black one is for clear conditions, the
red one is for cloudy conditions (both contaminated and overcast) and the blue one is for the whole
dataset. Linear regression is computed using the pyranometer values as $x$ and MSG-GHI forecast as
$y$. The parameters of the linear regressions are shown in the respective colours: $a$ is the slope, $b$ is
the intercept, $r$ is the correlation coefficient, $N$ is the number of data. The probability to have a
correlation coefficient larger than that found by chance is also shown ($p>r$). A small value of this
probability shows a high significance of the regression. The data for cloudy conditions of Figure 3a

show larger deviations from their regression line compared to clear sky data. This is confirmed by the correlation coefficient, which is 0.96 for clear sky and 0.89 for contaminated and overcast conditions. Also, the slope (intercept) of the linear regression is closer to 1.0 (closer to 0.0) for clear sky, in better agreement with the perfect regression.

Considering Figure 3a, two types of error are evident: a) there are cases when the cloud classification by MSG-GHI is wrong as, for example, for the black dots in the upper-left part of Figure 3a. For these points, the MSG-GHI is high (larger than 600 W/m$^2$) while the pyranometer observation is below 300 W/m$^2$. This error becomes particularly important for mountainous stations because, when the soil is covered by snow, it is more difficult for the MSG-GHI algorithm to correctly identify the clouds; b) the correlation coefficient for cloudy conditions is lower compared to clear sky data and shows the difficulty to correctly estimate the cloud optical depth, which can result in both overestimation of the MSG-GHI, i.e. the cloud optical depth is underestimated, or underestimation of the MSG-GHI, i.e. the cloud optical depth is overestimated. It is important to note that red points may also contain cases of wrong cloud classification. Nevertheless, the larger spread of the red points compared to the black ones shows, indirectly, the overall good classification of the sky conditions by MSG because the estimation of the GHI is more difficult for cloudy skies.

Figure 3b shows the scatter plot for the same station for the RAMS-GHI one-day hourly forecast. Linear regression is computed using the pyranometer hourly values as $x$ and corresponding RAMS-GHI forecast as $y$. The RAMS-GHI forecast data show larger deviations from their regression line compared to MSG-GHI. The correlation coefficient of the linear fit is 0.91 for clear conditions, while it is 0.60 for contaminated and overcast sky, showing a rather poor performance of the RAMS-GHI one-day hourly forecast in cloudy conditions. Both values are lower than the corresponding values of the MSG-GHI estimate.

Figure 3b for clear sky shows cases when RAMS predicts clouds that are not observed, i.e. the black dots in the lower right part of the figure, and cases when RAMS does not predict clouds that are observed, i.e. the red dots in the upper-left part of the figure. Also, the large deviations of the red dots from their regression line show either cases of incorrectly predicted sky conditions or errors in the representation of the cloud optical depth.

From Figure 3 it follows that: a) the performance in clear conditions is better compared to cloudy sky; b) the hourly estimate of the GHI by MSG outperforms the RAMS forecast. For the latter point, however, it is emphasized that the MSG and RAMS performance cannot be directly

compared because RAMS is a forecast, while MSG is an estimate of the GHI from radiance observations.

The results of Figure 3, even if shown for Vigna di Valle are found for all stations considered in this paper, and are similar to the findings of several studies (Kosmopulos et al., 2015; Lara-Fanego et al., 2012; Gomez et al., 2016).

To show this point for other stations, Figure 4 shows the RMSE as a function of the cloud coverage for MSG-GHI (Figure 4a) and for RAMS-GHI forecast (Figure 4b). In Figure 4a, the coloured bars for each sky condition (1=clear, 2=contaminated and 3=overcast) show the GHI average computed from the pyranometer hourly data, while the grey bars in the background show the RMSE of the MSG-GHI estimate for the different sky conditions for hourly data.

Figure 4a shows that the GHI decreases for the sky changing from clear to cloudy conditions, while the RMSE is higher when sky conditions become cloudier. More specifically, the RMSE is between 50 and 150 W/m$^2$, depending on the station, for clear sky, between 50 and 200 W/m$^2$ for contaminated conditions, and between 80 and 200 W/m$^2$ for overcast conditions.

Figure 4b shows the performance of the RAMS-GHI forecast as a function of the sky conditions. The values of the pyranometers are the same as in Figure 4a and are shown to help comparison. The RAMS-GHI one-day forecast RMSE increases from clear to overcast conditions and the error is higher compared to MSG-GHI. More specifically, excluding mountainous stations, which have larger errors, the RMSE is 100 W/m$^2$ for clear sky, 150-250 W/m$^2$, depending on the station, for contaminated sky, and around 250 W/m$^2$ for overcast conditions. In the latter case the RMSE is larger than the GHI for most stations, i.e. the relative error is larger than 100%.

Because of the dependence of the MSG-GHI estimation and RAMS-GHI forecast on the sky condition, a large variability of the performance is expected with the seasons and with the stations, because the cloud coverage at each site varies with the season and, for each season, from site to site. This point is investigated in the following sections.

### 3.2 Performance dependence on the season and cloud cover

Figure 5a shows the MBE of the MSG-GHI hourly estimate in all sky conditions for the different seasons, for the whole year and for all stations. Focusing on the whole year, there are five stations where the GHI is overestimated (maximum value at Monte Cimone; 18 W/m$^2$) and seven stations where the GHI is underestimated (minimum value at Pratica Di Mare; -12 W/m$^2$). The MBE is, however, small in absolute value and it is lower than 10 W/m$^2$ for seven pyranometers. Considering

the variability of the results with the station in all seasons, we note the larger absolute values for mountainous stations. This is expected because there are a larger number of cloudy data for those stations (Table 3) and the performance of the GHI estimate by MSG is worse for cloudy conditions (Figure 3a). This result is general and applies also to the RAMS forecast.

Figure 5b shows the MBE for the RAMS-GHI one-day hourly forecast. Considering the statistics for the whole year it is noted that the values are in general positive and below 30 W/m$^2$, with the exceptions of Paganella and Aosta where the MBE is negative, i.e. the RAMS forecast underestimates the GHI, and reaches the huge value of -120 W/m$^2$. The same behavior is found for all seasons, with few exceptions. Excluding the mountainous stations of Aosta and Paganella, the largest MBE is found in summer, showing the tendency of the RAMS forecast to overestimate the GHI in this season, while the smallest values occur in spring. Considering the dependence of the MBE with the station, it is evident the worse performance for mountainous stations, namely Paganella and Aosta, compared to maritime stations. The inspection of the model output for those stations reveals that the main source of error was the over forecast of cloudy conditions, as shown by the scatter plots between the RAMS-GHI one-day hourly forecast and the corresponding pyranometer values for these stations, given as a supplement to this paper. It is not easy to find the reason for this behaviour, because several factors could be involved as errors in the physical and numerical parameterizations of the model, and errors in the initial and boundary conditions. Also, the 4 km horizontal resolution is not enough to resolve the fine orographic structures over the Alps (Aosta and Paganella) and over the Apennines (Monte Cimone), and their interaction with the atmosphere.

Figure 6a shows the RMSE for the MSG-GHI hourly estimate in all sky conditions for different seasons, for the whole year and for the twelve stations. Considering the whole year, we note two groups of stations: the first with values around 100 W/m$^2$ containing the maritime and hilly stations, the second with values larger than 150 W/m$^2$ containing the mountainous stations. The increase of the RMSE for mountainous stations is caused mainly by: a) the 3*5km$^2$ horizontal resolution of the MSG-GHI can be not enough to represent the local sky conditions at the pyranometer, especially for mountainous stations where the complex orography determines rapid changes of the cloud coverage in short distances; b) The classification of sky conditions is more difficult where the soil is covered by snow and, because this condition is more frequent for mountainous stations, it increases the MSG-GHI error for those stations; c) The estimate of the hourly GHI by the MSG is more difficult in cloudy conditions (Figure 4), which are more frequent for mountainous stations. The different

performance of the two groups of stations is confirmed for all the seasons and highlights the
difficulty to clearly distinguish and classify clouds for the specific sites.
Considering the behavior of the RMSE with the season, the lowest values are often found in winter
even if the performance does not vary sizably with the season. Winter has also the lowest RMSE
averaged over all stations (84 W/m$^2$), followed by fall (98 W/m$^2$), summer (118 W/m$^2$), and spring
(125 W/m$^2$). The performance in winter is better compared to other seasons because the RMSE
statistic is sensitive to the larger errors (Wilks, 2006), and the departures of the GHI estimate from
the observation is lower in winter because the GHI is smaller. It is also noted the larger variability
of the performance in summer compared to other seasons, which will be discussed later on in this
section.
Another interesting statistic to quantify the performance of the MSG-GHI hourly estimate is the
rRMSE, which is shown in Table 4. Considering the whole year, this value ranges from 14% for
Cozzo Spadaro to 53% for Monte Cimone; for maritime and hilly stations the rRMSE is below
30%, while it is above 40% for mountainous stations, showing again the difference between the two
groups. The rRMSE has the smallest value in summer and the highest value in winter. While this
result is in part determined by the larger observed values of the GHI in summer, this analysis shows
more clearly the impact of the cloud coverage on the MSG-GHI performance. The percentage of
cloudy conditions is larger in winter compared to summer for all stations (Table 3) and the error of
the MSG-GHI is higher in cloudy conditions, as shown by the rRMSE. However, the larger
differences between the MSG-GHI hourly estimate and the pyranometer observation in summer,
even if in fewer occasions, determine larger values of the RMSE compared to winter, as shown in
Figure 6a.
Figure 6b shows the RMSE for the RAMS-GHI one-day hourly forecast. Considering the whole
year, the RMSE is below 200 W/m$^2$ for all stations with the exception of the mountainous stations,
where the error is larger because of the difficulty of the RAMS forecast to correctly predict the
cloud coverage. Considering the RMSE behavior for different seasons, averaged for all stations, the
lowest error is found in winter (142 W/m$^2$) followed by fall (171 W/m$^2$), summer (186 W/m$^2$) and
spring (245 W/m$^2$). Summer has the largest RMSE spread among the stations. In particular, it
shows the lowest error among all stations and seasons (Cozzo Spadaro, 110 W/m$^2$) but also values
larger than 300 W/m$^2$ for Paganella and Aosta. This result is caused by the large differences
between the RAMS-GHI one-day hourly forecast and observations. These differences are the
largest in summer (the lowest in winter) when the forecast of the cloud coverage is incorrect,

causing the largest (lowest) spread of the performance among stations. This applies also to the MSG-GHI hourly estimate.

The RMSE of the RAMS-GHI one-day hourly forecast is more than twice that of the MSG-GHI considering both the whole year and the seasons. The mountainous stations are an exception also in this case because the performance of MSG and RAMS are closer. A better performance of the MSG-GHI estimate is expected, because it is derived from the observations, while the RAMS is a forecast, however the results of this section quantify the difference between the two GHI sources in different conditions.

The rRMSE for the RAMS-GHI is shown in Table 5. Considering the yearly statistic, the values range from 31% for Cozzo Spadaro to 81% for Aosta. The rRMSE varies considerably between the mountainous stations compared to maritime and hilly stations, jumping from 53% obtained for Trieste (the worst performance for maritime and hilly stations) to 72% of Paganella (the best performance for mountainous stations). The variability of the rRMSE with the seasons shows again the important impact of the cloud coverage on the RAMS-GHI one-day hourly forecast performance. The smallest rRMSE are in summer, and the largest in winter for all stations. Moreover, for Trieste, Cimone and Aosta the rRMSE is about 100 % or larger in winter.

Before concluding this section, it is interesting to compare the RAMS-GHI one-day hourly forecast with the one-day hourly persistence forecast (Table 6). The one-day hourly persistence forecast was computed using hour by hour the observed values of the previous day.

 Considering the yearly statistics, the RAMS-GHI has a lower error compared to the one-day persistence forecast for all pyranometer but Paganella. The improvement given by RAMS is larger than 10% of the RMSE, showing a sizable impact. However, for Aosta, the difference between the two forecasts is negligible.

Considering the performance of the RAMS-GHI and one-day persistence hourly forecasts with the seasons, we note that: a) in winter the performance of the one-day persistence forecast is better than the RAMS-GHI forecast for seven pyranometers. This result is obtained for six stations in fall, four stations in spring and one station in summer; b) for mountainous stations the one-day persistence hourly forecast is better than the RAMS-GHI one-day hourly forecast for most-cases. These results show again the important impact of the cloud-coverage on the performance of the RAMS-GHI one-day hourly forecast, nevertheless the RAMS forecast can give added valued to the GHI forecast in most cases.

### 3.3 Daily evaluation and MOS application

In this section, we discuss the impact of the time interval on the RAMS-GHI and MSG-GHI performance.

Figure 7a shows the rRMSE for different stations and seasons for the RAMS-GHI one-day forecast. This figure is still computed from hourly data, as in the previous section (Figure 4b), but the RMSE is expressed in percentage to help comparison among statistics presented in this section.

Figure 7b shows the rRMSE for daily integrated GHI. Comparing the result of Figures 7a and 7b, it is apparent the impact of the time interval on the rRMSE. Considering the yearly result, for example, the rRMSE is reduced by more than 9% (in percentage units and the percentage is computed respect to the corresponding observations, Eqn. (2) and Eqn. (3)) for all stations when the statistics are computed for daily integrated GHI, and for several stations the improvement is larger than 15%. This improvement is found for all seasons and stations. In addition to the way used to compute the statistic, which produces smaller values compared to the same statistic for hourly data, the improvement is also caused by a partial compensation of the forecast underestimation and overestimation of the GHI during the day.

Considering the rRMSE for the MSG-GHI, a similar improvement is found, when computed for daily integrated GHI (Table 4). For the yearly statistics, the rRMSE decreases by 10% or more for all stations and an improvement larger than 5% is found in all seasons with a considerable variation among the stations.

### 3.4 MOS application

The last problem considered in this paper is the impact of the Model Output Statistics (MOS) on the one-day RAMS-GHI forecast and on the MSG-GHI, both for hourly and daily integrated GHI. The MOS was computed for each season and the "leave one" methodology was used to verify the RAMS forecast (MSG estimate) using MOS. This method is a cross-validation method to assess how the MOS prediction will perform in practice. For each hour of a season, the dataset is divided in two parts: a) the actual data (or actual value), which is the value at the selected hour of the RAMS one-day hourly forecast (or the MSG hourly estimate of GHI) and the corresponding pyranometer observation, and: b) the training dataset, which is composed by all data in the season with the exception of the actual data. The Eqn. (5) is computed for the training dataset ($y$ is the

pyranometer value and $x$ is the RAMS one-day hourly forecast or MSG hourly estimate of GHI), and it is applied to the actual data, which is the $x$, to give the corrected forecast ($y$). Because the MOS is computed starting from hourly data, the training period is all the season but one hour. This procedure was repeated for all the hourly data in the season, obtaining the time series of the corrected RAMS one-day hourly forecast and of the corrected MSG hourly estimation of the GHI. The RMSE and rRMSE were computed for the corrected forecast/estimate of the GHI. In this way, the data used for computing MOS is statistically independent from the dataset used for the verification.

The statistic computed from hourly data are shown in Table 6 for the RAMS forecast. It is apparent that the MOS improves the RAMS performance especially for Aosta and Paganella, where the Bias is high (Figure 5b). In particular, after the MOS application, the absolute value of the Bias is less than 30 W/m$^2$ for Paganella and Aosta for all seasons as well as for the whole year (not shown). With the MOS application, the RAMS-GHI one-day hourly forecast performs better than the one-day persistence hourly forecast for all stations considering the whole year, even if there are still occasions when the one-day persistence hourly forecast has a better performance than the RAMS-GHI one-day hourly forecast (Paganella in winter and fall, Aosta in winter, spring and fall, Trapani in winter).

Starting from hourly data after the MOS correction, daily integrated GHI statistics were also computed. The rRMSE of RAMS-GHI one-day forecast is shown in Figure 7c and Table 5. The rRMSE decreases by 2-8% (in percentage units) for most stations compared to the daily integrated GHI without MOS, with exception of Paganella and Aosta, where the improvement is larger. This is expected because the Bias is larger for these stations (Figure 5b) and the MOS is a technique that improves the forecast by reducing the Bias. This is confirmed by the inspection of the rMBE (not shown), which is reduced by the application of the MOS.

The application of the MOS to the MSG-GHI gives no improvement on both rRMSE (Table 4) and rMBE (not shown). This is caused by the small values of the Bias of the MSG-GHI (Figure 5a).

4. **Summary and conclusions**

In this paper, we analyzed the performance of the MSG-GHI estimation and RAMS-GHI one-day forecast for one year (1 June 2013 - 31 May 2014) over the Italian territory. Twelve pyranometers, scattered over the country and representing a variety of climate characteristics, were used to

evaluate the performance. The analysis was performed for both hourly values and daily integrated GHI, and the dependence with the season and sky conditions was studied.

The results for the analysis on hourly data show the dependence of the MSG-GHI estimation and RAMS-GHI forecast on the sky conditions, which mirrors in a notable dependency with the season and station. In particular, mountainous stations have worse performance compared to hilly and maritime stations.

The analysis of the MBE for the RAMS-GHI shows that the one-day hourly forecast overestimates the GHI, with the exception of the mountainous stations of Paganella and Aosta, where a considerable underestimation is found. The MSG-GHI doesn't show a specific behavior of the MBE with both overestimation and underestimation, depending on the season and station.

The RMSE for the RAMS-GHI one-day hourly forecast is the lowest in winter, followed by fall and spring. In summer, the RMSE shows the largest difference among the stations, the maritime stations showing the best performance, because the RMSE is sensitive to the departures between forecast and observation, which are larger in summer when the cloud coverage is not well predicted or estimated at the site.

The RMSE of the MSG-GHI hourly estimate is more than halved compared to RAMS-GHI, with the exception of the mountainous stations where the RMSE of the two datasets are closer.

The cloud coverage has an important impact also on the RMSE of both MSG-GHI hourly estimate and RAMS-GHI one-day hourly forecast. The error is higher for cloudy conditions compared to clear sky. This is especially evident for RAMS because the RMSE averaged over all the stations varies from 91 $W/m^2$, to 191 $W/m^2$, and to 245 $W/m^2$ for clear, contaminated and overcast conditions, respectively; for MSG-GHI, the RMSE averaged over all stations varies from 68 $W/m^2$, to 123 $W/m^2$, and to 98 $W/m^2$ for clear, contaminated and overcast conditions, respectively. However, the analysis of the rRMSE reveals more clearly the impact of the cloud coverage on the performance. Both RAMS-GHI one-day hourly forecast and MSG-GHI hourly estimate show the largest rRMSE in winter and the lowest in summer, following the behaviour of the cloud coverage.

The increase of the RMSE with the cloud coverage is a combination of both the inability of the two methods to correctly represent the cloud coverage and of the difficulty to compute the GHI in cloudy conditions.

The results for daily integrated GHI show a notable improvement of the RAMS-GHI and MSG-GHI performance. The partial compensation of overestimation and underestimation during the day

improves the performance for the daily integrated GHI. This result is similarly shown in other
studies for different countries (Lara-Fanego et al., 2012; Kosmopulos et al., 2015; Gómez et al.,
510 2016).

Applying a simple post-processing technique, i.e. the MOS, to the RAMS-GHI one-day hourly
forecast reduces the RMSE (2-8% of its value), while the MOS has a negligible impact on the
MSG-GHI RMSE.
The performance of the RAMS-GHI one-day hourly forecast, with and without the MOS correction,
has been compared with the one-day persistence hourly forecast to quantify the added value of the
RAMS forecast. The results show that the RAMS forecast, especially with the MOS correction,
outperforms the one-day persistence forecast and that the improvement is often larger than 10% of
the RMSE. Nevertheless, there are still few occasions (Paganella in winter and fall, Aosta in winter,
spring and fall, and Trapani in winter) when the one-day persistence forecast outperforms the
RAMS forecast.
The results of this paper are representative of the current operational implementation of the RAMS
model at ISAC-CNR. There have been recent improvements to the RAMS model (CSU-RAMS,
http://vandenheever.atmos.colostate.edu/vdhpage/rams.php) that will be explored in future studies
to improve the GHI forecast. The errors of the RAMS forecast for the GHI can be divided in three,
non-exhaustive, main components: a) errors in the prediction of the cloud coverage; b) errors in the
simulation of the interaction between the radiation and the clouds; c) errors in the representation of
the aerosol effects on the GHI.
As shown by the results of this and others papers, the error (RMSE) on the prediction of the GHI is
of the order of the GHI when the cloud coverage is not well represented. Errors of both physical and
numerical parameterizations of the model, but also errors in the initial and boundary conditions
contribute to this issue. In particular, the microphysical scheme influences the whole simulation
through a multitude of dynamic, radiative, thermodynamic and microphysics processes. The WSM6
scheme used in this paper is a single-moment scheme, predicting the mixing ratios of six
hydrometeors (vapour, cloud, rain, graupel, ice, snow). The WSM6 gave better performance
compared to other single-moment microphysics schemes included in RAMS for twenty cases over
Italy characterized by widespread convection and, for this reason, it is used in the operational
implementation at ISAC-CNR. However, the inability of single-moment schemes to allow the
number concentration and mean diameter of hydrometeors to vary independently limits their ability
to simulate clouds with characteristics consistent with observations across a wide range of

atmospheric conditions. Also, the sensitivity of these schemes to fixed parameters as, for example, the number concentration of the hydrometeors, is high (Igel et al., 2015).

When both the mixing-ratio and number concentration can be predicted, as in double-moment schemes, the description of the physical processes as condensation, collision-coalescence, and sedimentation is improved. For this reason, double-moment schemes outperform single-moment schemes as shown in several studies (Igel et al., 2015 and references therein).

The CSU-RAMS model includes a double-moment microphysics scheme (Meyers et al., 1997) that could improve the prediction of the cloud coverage and will be considered in future studies.

Also, the cumulus parameterization scheme has an important role on the NWP forecast, especially for cloud prediction. In addition to the Kuo scheme, used in this paper for the first domain, RAMS implements the Kain-Fritsch scheme (Castro et al., 2005). This scheme will be used in future studies to assess the sensitivity of the performance to the choice of the cumulus parameterization scheme.

Another important point to consider for improving the model performance of the GHI forecast is the change in the optical properties of the clouds when the liquid and ice phases are considered in the radiative scheme (Harrington et Olsson, 2001; Sun and Shine, 1995). The Chen and Cotton scheme (Chen and Cotton, 1983) used in this paper, while fast and efficient from the computational point of view, considers the total condensate in the atmosphere but not the phase of the water (i.e. ice, liquid or mixed). Numerical and observational experiments (Harrington et Olsson, 2001; Sun and Shine, 1995) show that the impact of the water phase is significant for the computation of the GHI because the absorption and emissions are largely reduced in ice compared to liquid path with the same water path.

Finally, our radiative scheme neglects the impact of the aerosols. This impact, however, can be very important. For example, Lara-Fanego et al. (2012) show that the overestimation of the GHI by WRF over Andalucia in clear sky conditions was caused by the underestimation of the aerosol optical depth (AOD), which was assumed 0.1 for their experiments. Zamora et al. (2005) showed that a doubling of the AOD considered in the Dudhia scheme (Dudhia, 1989) was responsible for a decrease of the GHI of about 100 $W/m^2$ at the solar noon over US. Kosmopulos et al. (2017) investigates the impact of an extremely high dust event (maximum AOD 3.5), occurred from 30 January to 3 February 2015 over Greece. For this event, they found an attenuation of the GHI up to 40-50 %. They also show that, for climatological conditions, the attenuation of the GHI by the aerosol load is less than 10%. Considering the above results and the fact that the RMSE statistic

used in this paper is sensitive to large errors, an important impact of the aerosols is expected. The Harrington et al. (1997) radiation scheme is aerosol sensitive, is available in CSU-RAMS, and will be tested in future studies.

To put the results of this paper in a more general context, we compare our statistics with similar studies in the Mediterranean area (Greece and Spain).

Kosmopulos et al. (2015) quantified the performance of the MM5 model for the one- and two-days forecast over Greece. The forecast was compared with eleven pyranometers displaced over the country. The RMSE computed from hourly data and for the one-day forecast ranges between 160 $W/m^2$ for the Chania station to 230 $W/m^2$ for Amfiklia. The error increases with the terrain complexity and cloud coverage: Chania is located in the western part of the Crete Island and shows a Mediterranean climate, while Amfiklia is located in one of the highest plateaus of Greece, bounded at the west by the Pindos mountain. The RMSE shows a small increase between the first and second day of forecast. With the exception of the mountainous stations of this paper, where the RMSE is larger, our performance is in line with that of Kosmopulos et al. (2015). Also, both studies show a positive MBE with values of few tens of $W/m^2$ for most stations, with the exception of Paganella and Aosta stations of this study where the MBE is larger in absolute value.

Gómez et al. (2016) quantified the performance of the RAMS model (both versions 4.4 and 6.0) for the one-, two- and three-days GHI forecast over the Valencia Region. They considered thirteen pyranometers widespread over the region. Focusing on the RMSE for hourly data in summer, they found errors of 200 $W/m^2$ for flat terrain and 250 $W/m^2$ for hilly terrain. The RMSE for winter is 150-160 $W/m^2$, depending on the stations. The MBE is of few tens of $W/m^2$ and it is positive. They found similar results among the three days of forecast and also between the two versions of the RAMS model. With the exceptions of the mountainous stations of this paper, where both the RMSE and MBE in absolute value are larger, our results are in line with those of Gómez et al. (2016).

Lara Fanego et al. (2012) examined the performance of the WRF model for the GHI one- two- and three-days forecast over Andalucia (Spain). They consider four stations: Andasol, Jerez, Cordoba and Huelva. The RMSE computed from hourly data for the whole year is 140 $W/m^2$ for Cordoba, Jerez ad Huelva stations and 170 $W/m^2$ for Andasol. Differences of the RMSE among the three days of forecast are small. The RMSE of Lara Fanego et al. (2012) is smaller (10-20 $W/m^2$) than those of this paper. This result can be caused by the difference of the climate and orography at the stations considered in the two studies, nevertheless a better treatment of the interaction between aerosols and radiation in Lara Fanego et al. (2012) contribute to this difference. The MBE of Lara

Fanego et al (2012) is in line with that of this paper, with the exception of Paganella and Aosta stations.

Overall, the results of this paper show that the MSG-GHI estimate and the RAMS forecast have still big issues in cloudy conditions. In particular, considering the potential of the RAMS forecast to participate to the energy market, it is difficult to assess its usefulness from the results of this paper. While the RAMS forecast outperforms the one-day persistence forecast in clear sky, it has large errors in cloudy conditions and it is not easy to give a final balance between the advantages in clear conditions and disadvantages in cloudy conditions. Considering also the variability of the RAMS performance from site to site, the usefulness of the RAMS forecast from an economic perspective must be evaluated from case to case (Wittman et al. 2008).

**Acknowledgments**

The ECMWF and CNMCA (Centro Nazionale di Meteorologia e Climatologia Aeronautica) are acknowledged for the use of the MARS (Meteorological Archive and Retrieval System). Two anonymous reviewers and Stephen Saleeby are acknowledged for their comments that improved the quality of the paper.

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

730

**Tables and Figures**

Table 1: RAMS grid-setting for the first and second grids. NNXP, NNYP and NNZP are the number of grid points in the west-east, north-south, and vertical directions. Lx(km), Ly(km), Lz(m) are the domain extensions in the west-east, north-south, and vertical directions. DX(km) and DY(km) are the horizontal grid resolutions in the west-east and north-south directions. CENTLON and CENTLAT are the geographical coordinates of the grid centres.

|  | First grid | Second grid |
| --- | --- | --- |
| NNXP | 231 | 401 |
| NNYP | 231 | 401 |
| NNZP | 36 | 36 |
| Lx | 2772 km | 1600 km |
| Ly | 2772 km | 1600 km |
| Lz | ≈22 km | ≈22 km |
| DX | 12 km | 4 km |
| DY | 12 km | 4 km |
| CENTLAT (°) | 42.0 | 42.0 |
| CENTLON (°) | 12.5 | 12.5 |

Table 2: Station names, abbreviations, coordinates, height above the sea level (meters, forth column), instrument type and managing institution for the twelve sites.

| Station name | Abbreviation | Coordinates (lon;lat) | Height (m) a.s.l | Pyranometer type | Institution |
| --- | --- | --- | --- | --- | --- |
| Trapani | tra | 12.5; 37.9 | 9 | CM11 Kipp&Zonen | Aeronautica Militare |
| Cozzo Spadaro | csp | 15.1; 36.7 | 51 | CM11 Kipp&Zonen | Aeronautica Militare |
| Santa Maria di Leuca | sml | 18.3; 39.8 | 112 | CM11 Kipp&Zonen | Aeronautica Militare |
| Capo Palinuro | pal | 15.3; 40.0 | 185 | CM11 Kipp&Zonen | Aeronautica Militare |
| Pratica di Mare | pdm | 12.5; 41.7 | 32 | CM11 Kipp&Zonen | Aeronautica Militare |

| | | | | | |
|---|---|---|---|---|---|
| Vigna di Valle | vdv | 12.2; 42.1 | 266 | CM11 Kipp&Zonen | Aeronautica Militare |
| Pisa | pis | 10.4; 43.7 | 6 | CM11 Kipp&Zonen | Aeronautica Militare |
| Cervia | cer | 12.3; 44.2 | 10 | CM11 Kipp&Zonen | Aeronautica Militare |
| Trieste | tri | 13.8; 45.7 | 4 | CM11 Kipp&Zonen | Aeronautica Militare |
| Monte Cimone | cim | 10.7; 44.2 | 2173 | CM11 Kipp&Zonen | Aeronautica Militare |
| Paganella | pag | 11.0; 46.2 | 2129 | CM11 Kipp&Zonen | Aeronautica Militare |
| Aosta | aos | 7.4; 45.7 | 583 | CMP21 Kipp&Zonen | Arpa Valle D'Aosta |

752

753

Table 3: Percentage of data in clear, contaminated and overcast conditions for all stations and seasons, as well as for the whole year, estimated by CPP (Section 2.1).

| Station | Winter [%] | Spring [%] | Summer [%] | Fall [%] | Year [%] |
|---|---|---|---|---|---|
| tra | 48;23;29 | / | 82;15;03 | 38;39;23 | 60;24;16 |
| csp | 13;34;53 | 46;19;35 | 69;22;09 | 34;31;35 | 44;26;30 |
| sml | 33;31;36 | 37;40;23 | 62;31;07 | 41;37;22 | 44;34;20 |
| pal | 03;28;69 | 13;30;57 | 49;37;14 | 23;34;43 | 25;33;42 |
| pdm | 36;27;37 | 37;44;19 | 79;14;07 | 51;27;22 | 54;27;19 |
| vdv | 37;25;38 | 27;45;28 | 73;20;07 | 48;29;23 | 51;28;21 |
| pis | 34;22;45 | 38;33;29 | 77;16;07 | 44;29;27 | 52;24;24 |
| cer | 33;20;47 | 41;27;32 | 74;16;10 | 39;25;36 | 49;22;29 |
| tri | 20;21;59 | 31;29;40 | 64;24;12 | 34;23;43 | 42;24;34 |
| cim | 05;50;45 | 09;46;45 | 34;49;17 | 21;36;43 | 20;45;35 |
| pag | 23;22;55 | 39;27;34 | 45;38;17 | 27;31;42 | 35;31;34 |
| aos | 12;39;49 | 25;35;40 | 32;38;30 | 25;38;37 | 23;37;40 |

756

757

Table 4: rRMSE [%] for the MSG-GHI estimate computed for hourly and daily integrated GHI for different seasons and stations. The first number in each cell is the rRMSE computed using hourly data, the second number is the rRMSE computed for daily integrated GHI, the third number is the rRMSE computed after the MOS correction for daily integrated GHI (see text for details).

| Station | Winter [%] | Spring [%] | Summer [%] | Fall [%] | Year [%] |
|---|---|---|---|---|---|
| tra | 30; 3; 3 | / | 11; 4; 4 | 27; 5; 6 | 18; 6; 7 |
| csp | 20; 5; 3 | 14; 4; 4 | 9; 4; 3 | 19; 3; 6 | 14; 6; 6 |
| sml | 27; 4; 4 | 21; 6; 6 | 14; 5; 4 | 23; 6; 7 | 19; 8; 8 |
| pal | 25; 4; 3 | 20; 5; 5 | 11; 4; 4 | 39; 4; 5 | 23; 7; 7 |

| | | | | | |
|---|---|---|---|---|---|
| pdm | 28; 3; 3 | 17; 5; 5 | 12; 4; 4 | 19; 6; 7 | 17; 7; 7 |
| vdv | 27; 3; 3 | 24; 5; 5 | 18; 6; 6 | 24; 4; 6 | 21; 8; 8 |
| pis | 26; 4; 3 | 22; 6; 5 | 16; 6; 5 | 20; 4; 5 | 19; 7; 7 |
| cer | 27; 4; 4 | 21; 6; 5 | 15; 6; 5 | 23; 3; 6 | 20; 8; 8 |
| tri | 34; 3; 3 | 28; 6; 6 | 22; 9; 8 | 25; 5; 7 | 26; 10; 10 |
| cim | 92; 18; 19 | 60; 24; 27 | 43; 23; 21 | 47; 13; 17 | 53; 27; 28 |
| pag | 57; 12; 10 | 35; 17; 16 | 38; 17; 17 | 43; 12; 11 | 40; 21; 20 |
| aos | 89; 7; 10 | 43; 12; 9 | 44; 12; 17 | 53; 6; 9 | 51; 15; 17 |

Table 5: rRMSE [%] for the RAMS-GHI one-day forecast computed for hourly and daily integrated GHI for different seasons and stations. The first number in each cell is the rRMSE computed using hourly data, the second number is the rRMSE computed for daily integrated GHI, the third number is the rRMSE computed after the MOS correction for daily integrated GHI (see text for details).

| Station | Winter [%] | Spring [%] | Summer [%] | Fall [%] | Year [%] |
|---|---|---|---|---|---|
| tra | 58; 12; 8 | / | 20; 12; 10 | 49; 17; 17 | 33; 21; 19 |
| csp | 43; 12; 9 | 38; 23; 19 | 19; 11; 10 | 42; 15; 16 | 31; 22; 19 |
| sml | 57; 14; 11 | 47; 25; 19 | 26; 16; 12 | 42; 15; 13 | 38; 27; 21 |
| pal | 58; 16; 9 | 54; 25; 20 | 27; 18; 16 | 47; 16; 16 | 41; 28; 25 |
| pdm | 60; 14; 11 | 48; 28; 21 | 25; 15; 14 | 40; 12; 13 | 37; 27; 22 |
| vdv | 66; 14; 10 | 57; 28; 19 | 32; 19; 16 | 49; 14; 14 | 42; 29; 23 |
| pis | 68; 15; 10 | 56; 28; 21 | 32; 22; 18 | 51; 17; 17 | 45; 30; 25 |
| cer | 68; 13; 10 | 52; 26; 19 | 34; 20; 16 | 53; 14; 13 | 44; 29; 23 |
| tri | 97; 16; 11 | 63; 26; 19 | 44; 26; 20 | 58; 16; 15 | 53; 35; 27 |
| cim | 117; 22; 22 | 96; 44; 44 | 60; 39; 30 | 74; 24; 24 | 75; 48; 44 |
| pag | 86; 15; 10 | 77; 50; 28 | 66; 44; 26 | 79; 30; 30 | 72; 56; 36 |
| aos | 113;17; 17 | 78; 49; 25 | 71; 48; 43 | 84; 23; 23 | 81; 60; 42 |

Table 6: RMSE [W/m$^2$] for the RAMS-GHI one-day hourly forecast (first number in each cell), one-day persistence hourly forecast (second number in each cell) and RAMS-GHI one-day hourly forecast after the MOS application for different seasons and stations (third number in each cell, see text for details). Bold style shows the cases when the RAMS-GHI one-day hurly forecast has a worse performance compared to the one-day persistence hourly forecast.

| Station | Winter [W/m$^2$] | Spring [W/m$^2$] | Summer [W/m$^2$] | Fall [W/m$^2$] | Year [W/m$^2$] |
|---|---|---|---|---|---|
| tra | **149**; 120; 130 | / | 111; 136; 104 | **177**; 162; 163 | 152; 190; 139 |
| csp | 137; 169; 126 | 199; 218; 184 | 107; 168; 102 | 168; 191; 157 | 161; 204; 148 |

| | | | | |
|---|---|---|---|---|
| sml | 151; 170; 133 | 218; 275; 200 | 142; 178; 128 | 159; 186; 147 | 178; 236; 160 |
| pal | 138; 177; 125 | 232; 257; 212 | 145; 181; 141 | 173; 192; 161 | 186; 229; 171 |
| pdm | 140; 151; 123 | 226; 231; 206 | 133; 172; 132 | 144; 167; 139 | 176; 209; 161 |
| vdv | 138; 161; 115 | 230; 238; 196 | 168; 189; 158 | 158; 170; 140 | 182; 209; 159 |
| pis | **125**; 119; 104 | **227**; 223; 200 | 165; 180; 153 | 163; 174; 150 | 188; 216; 166 |
| cer | **120**; 118; 100 | 204; 241; 182 | 170; 206; 158 | **149**; 147; 139 | 178; 220; 157 |
| tri | **131**; 77; 181 | **207**; 195; 181 | 206; 223; 189 | **147**; 142; 134 | 190; 220; 166 |
| cim | **158**; 145; 160 | 288; 289; 288 | 253; 274; 220 | **199**; 193; 183 | 253; 293; 238 |
| pag | **148**; 95; 114 | **318**; 266; 239 | **304**; 291; 255 | **224**; 156; 183 | **286**; 276; 221 |
| aos | **172**; 99; 148 | **341**; 234; 256 | 326; 347; 281 | **200**; 126; 176 | 287; 294; 229 |


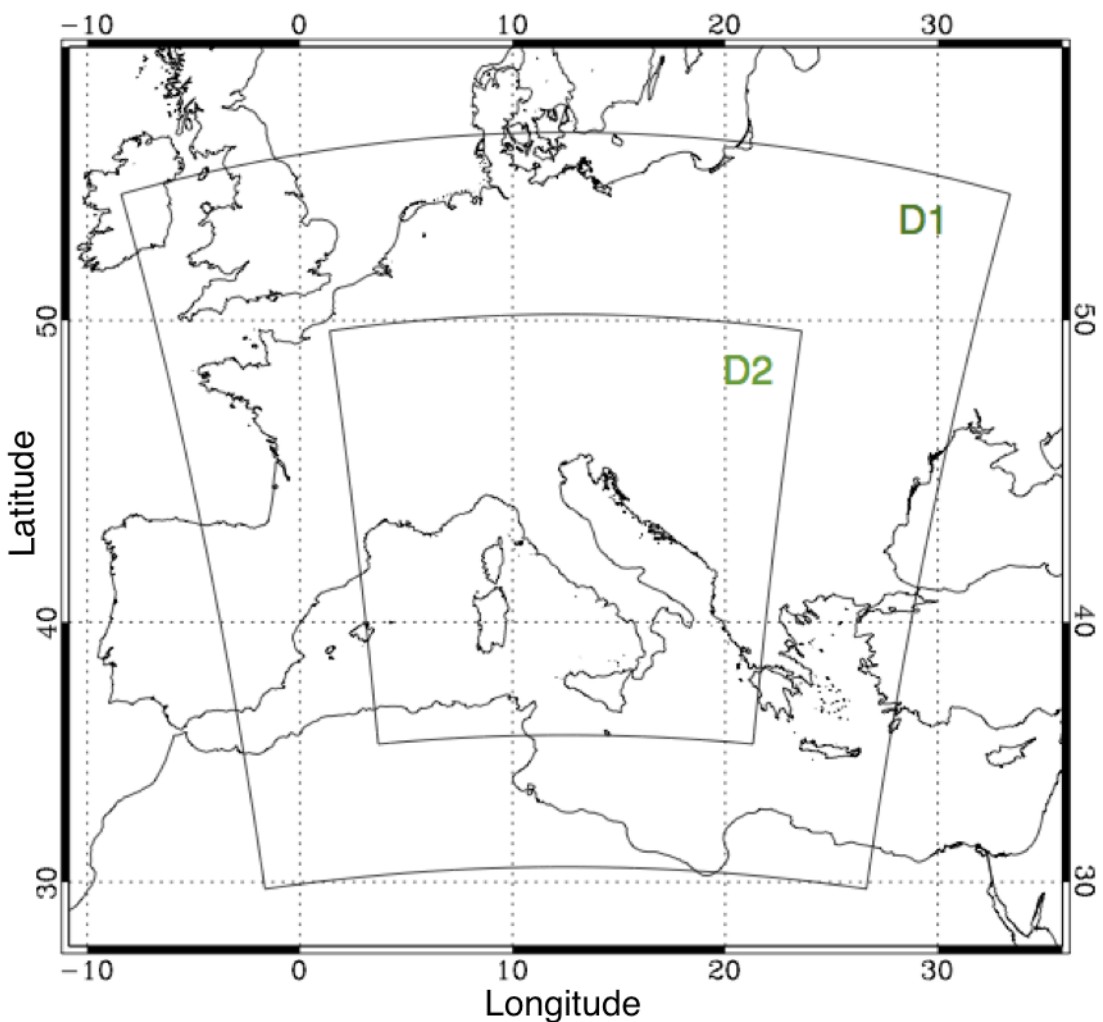


**Figure 1:** Model domains. The second domain has 4 km horizontal resolution and it is nested in the first domain, at 12 km horizontal resolution, using one-way nesting.


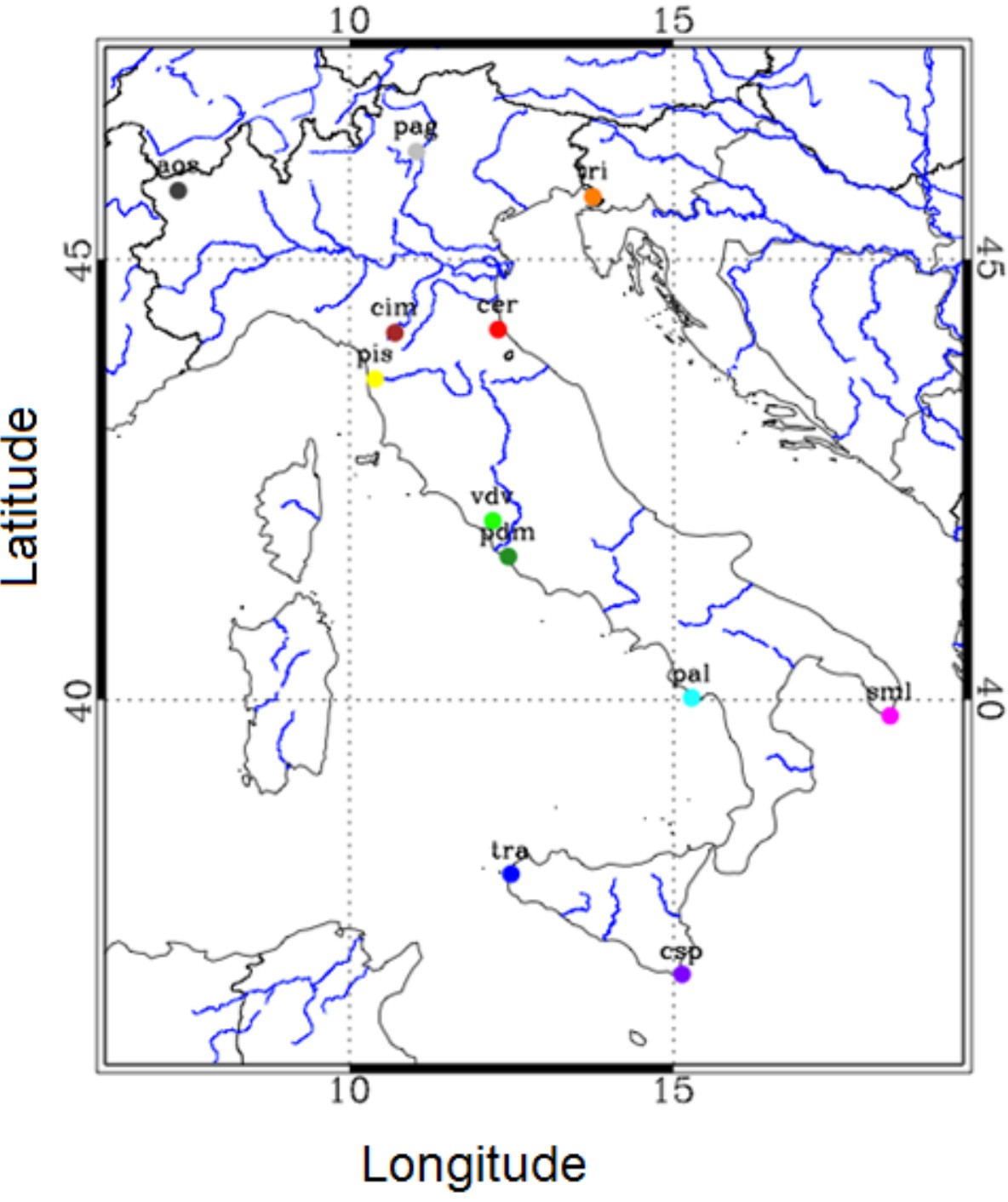


**Figure 2:** Stations distribution over the Italian territory.




**788**    **a)**

**789**

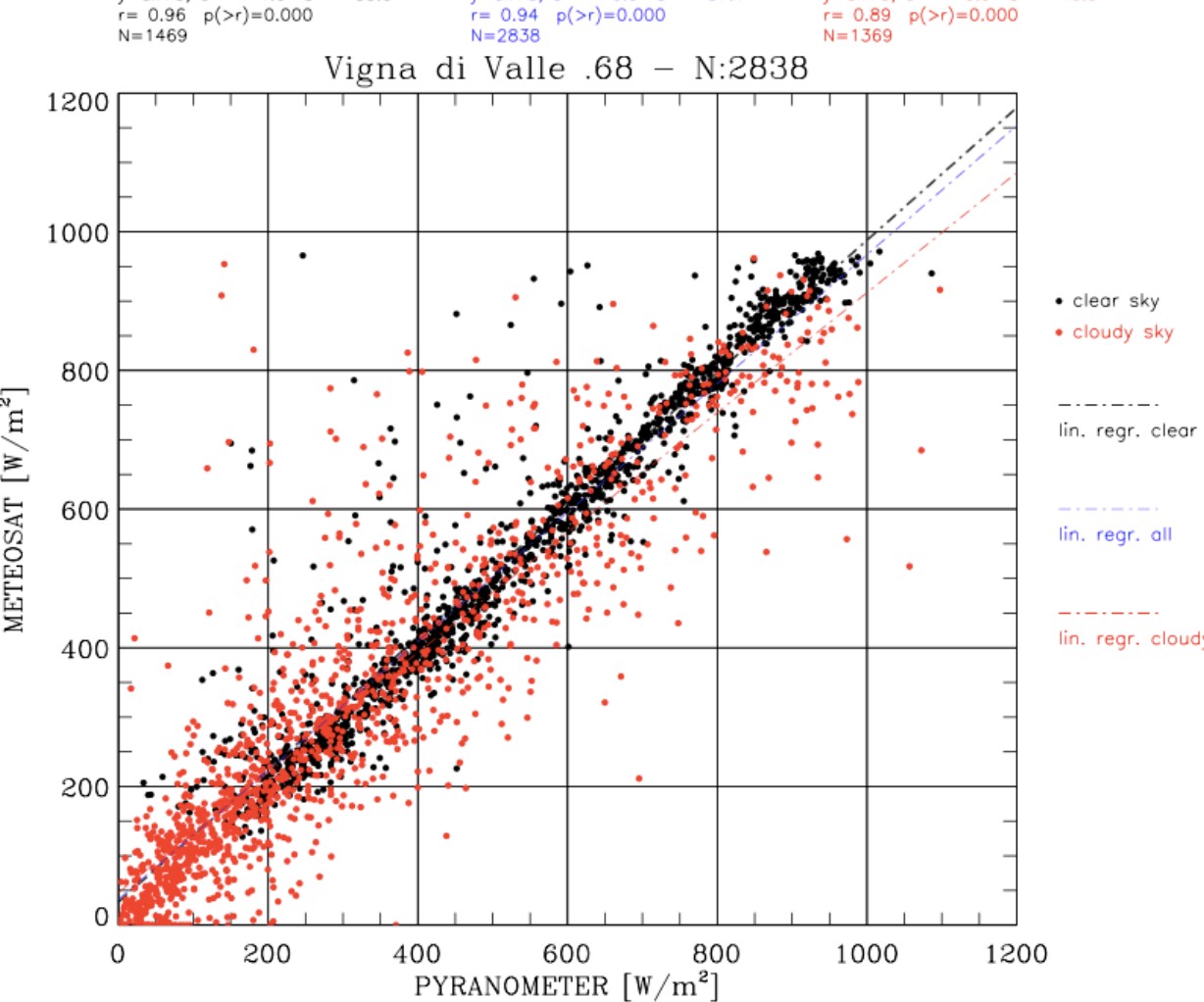

**790**

**791**

**792**

**793**

**794**

**795**

**796**

**797**

**798**

**799**

**800**

**b)**

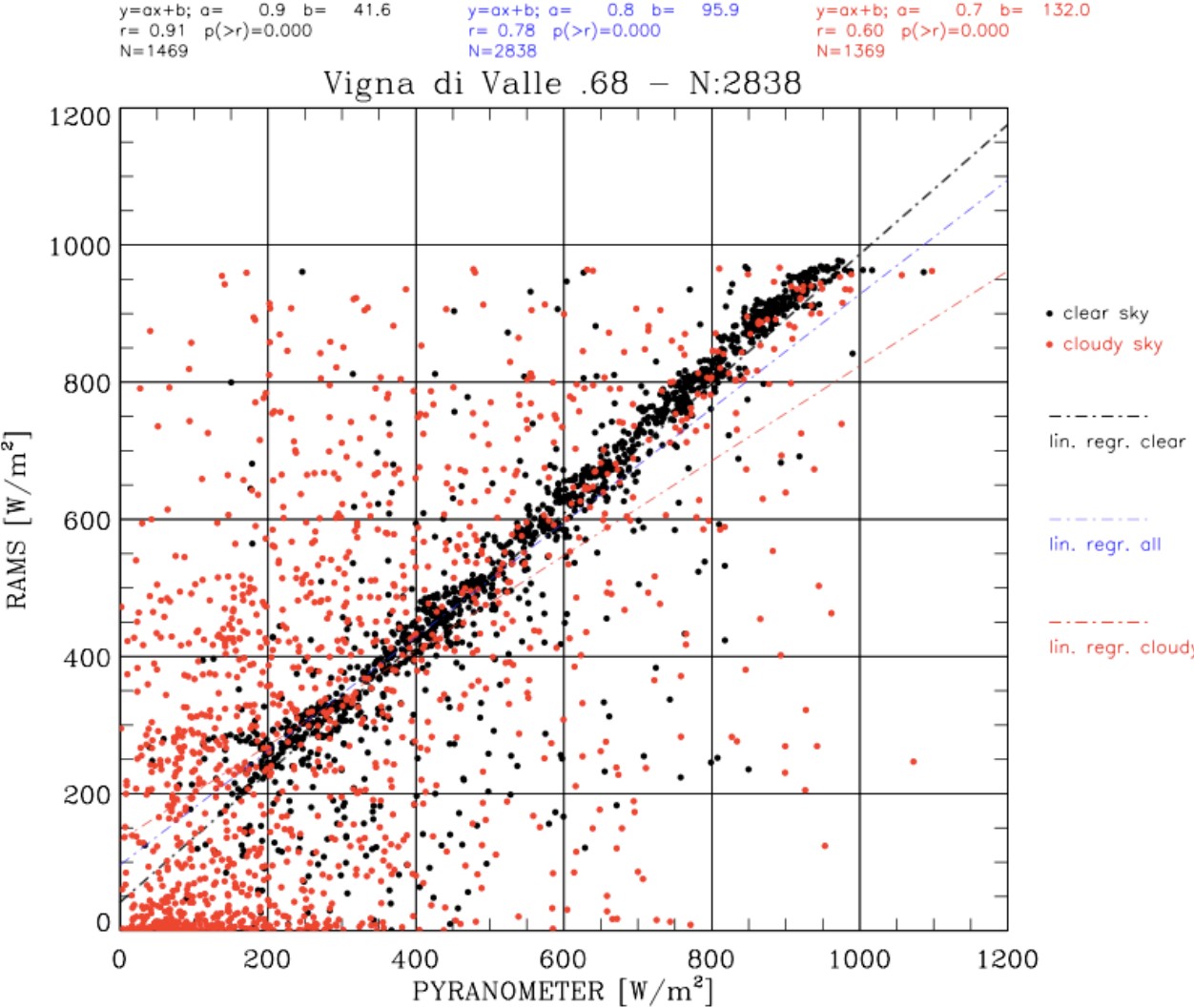



Figure 3: a) scatter plot of the GHI for the pyranometer (*x*-axis) and MSG (*y*-axis) hourly data. The
black dots are for clear sky conditions while the red dots are for both contaminated and overcast
skies; b) as in a) for the RAMS one-day hourly forecast. Regression lines are shown in their
respective colours (blue is for all data, i.e. both clear and cloudy conditions).







**a)**

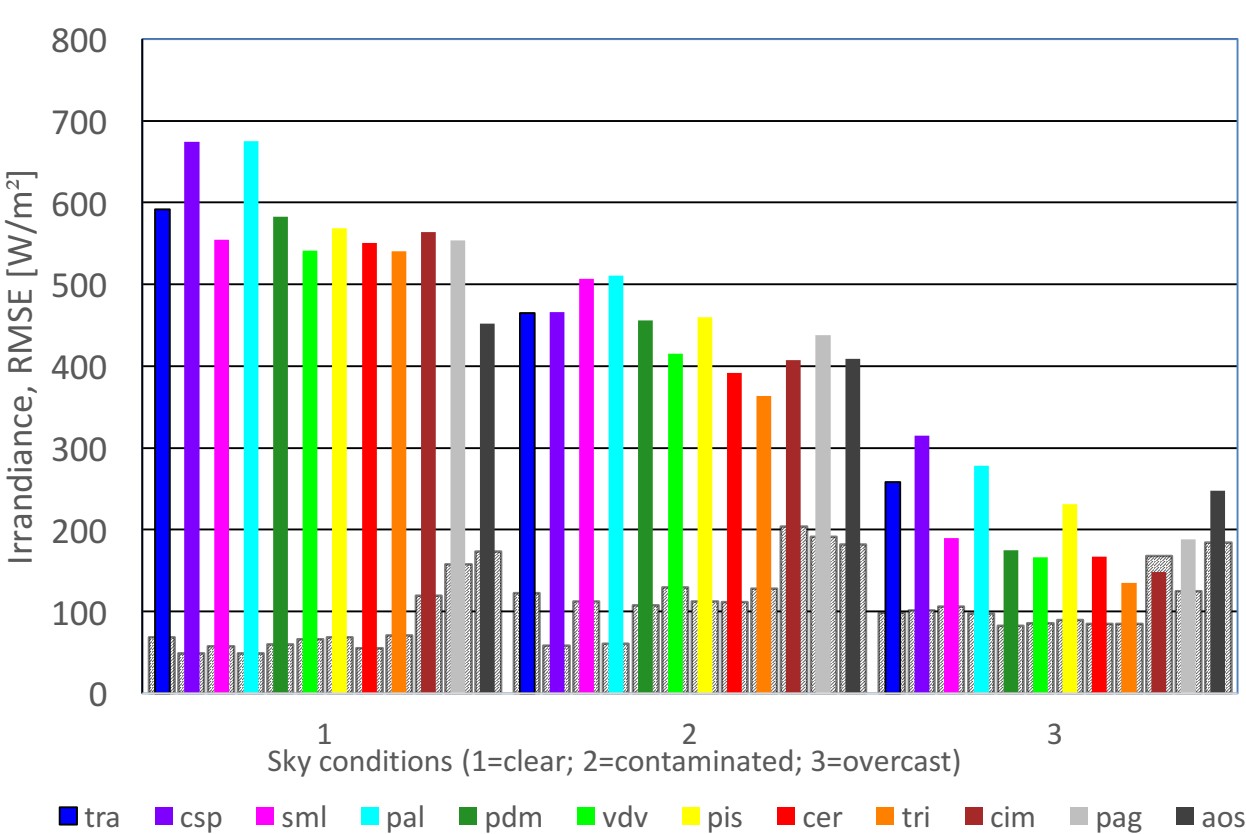













**b)**

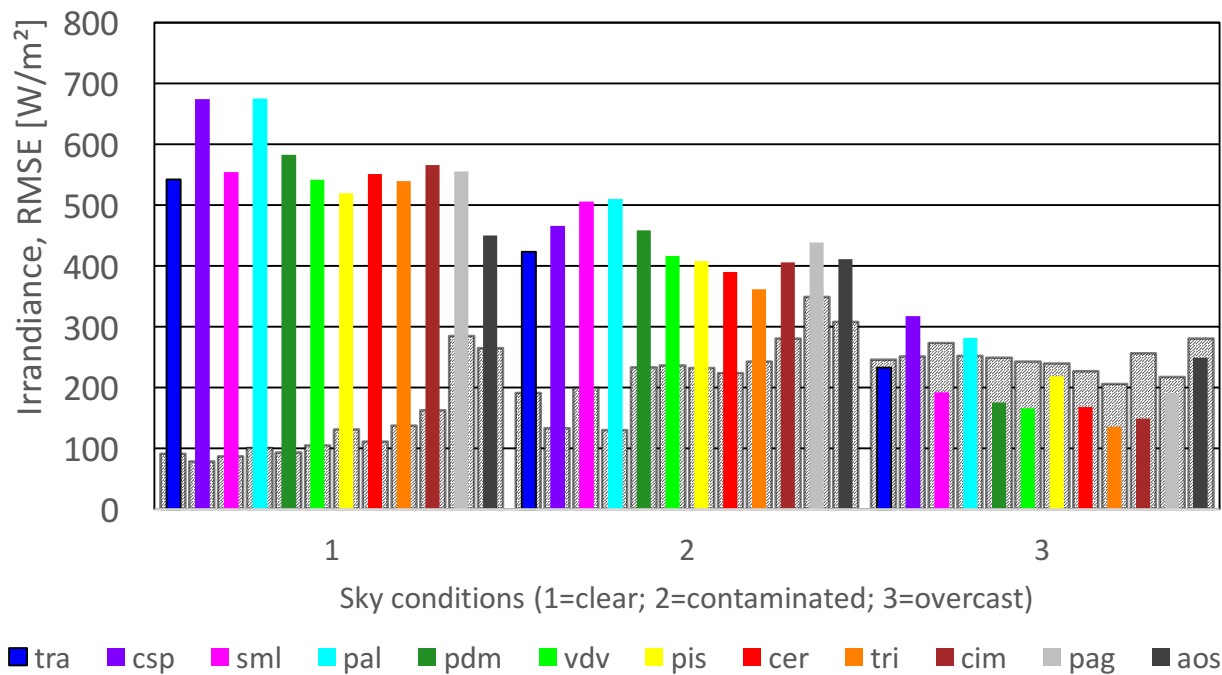

Figure 4: a) Mean irradiance (coloured bars) and RMSE (grey bars) for different sky conditions: clear (1), contaminated (2) and overcast (3) for the MSG-GHI estimate. The figure has been derived from the hourly data of pyranometers and MSG-GHI estimate. The RMSE is shown with the same scale as the mean irradiance; b) As in a) for the RAMS-GHI one-day hourly forecast.

846    **a)**

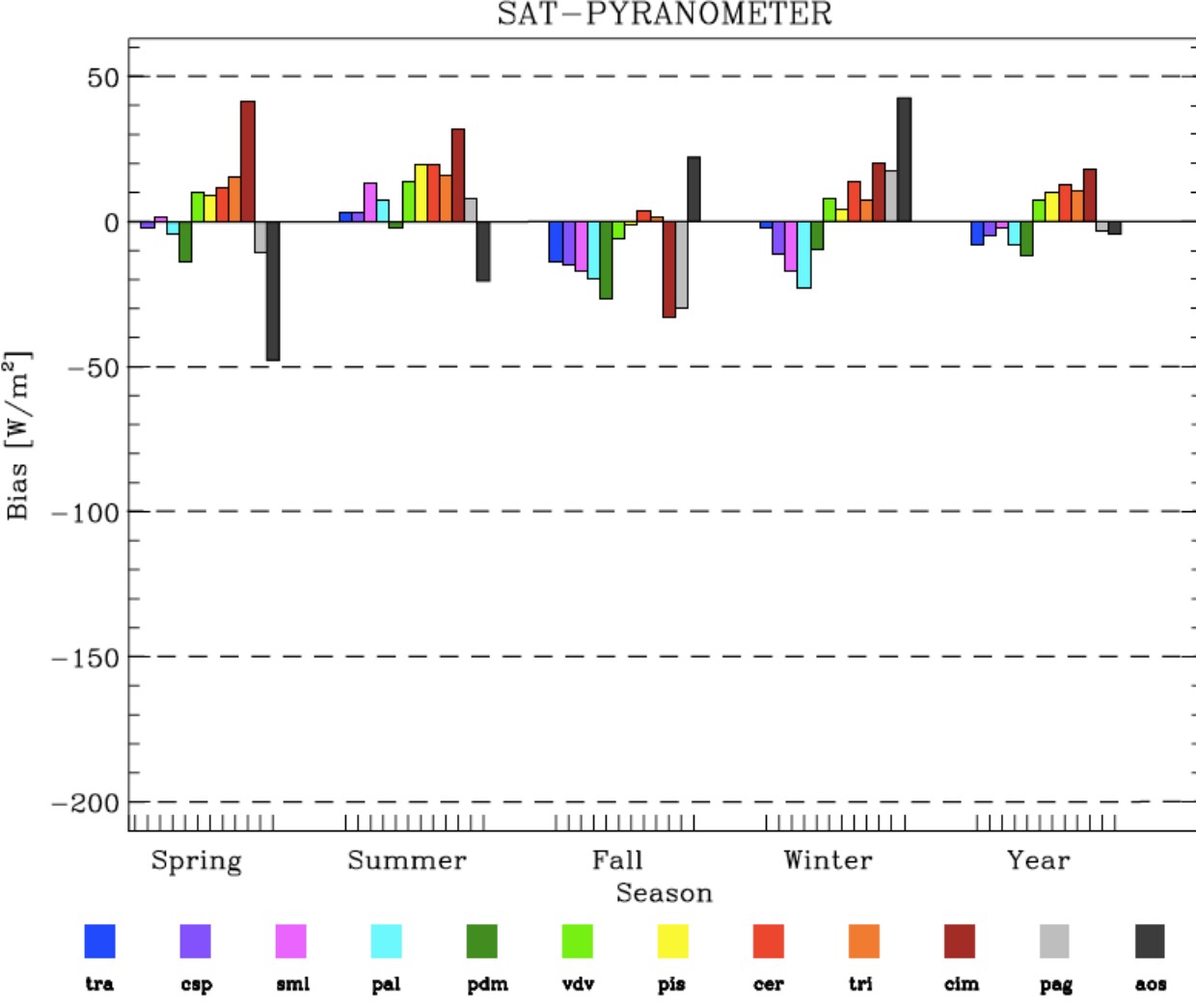

**b)**

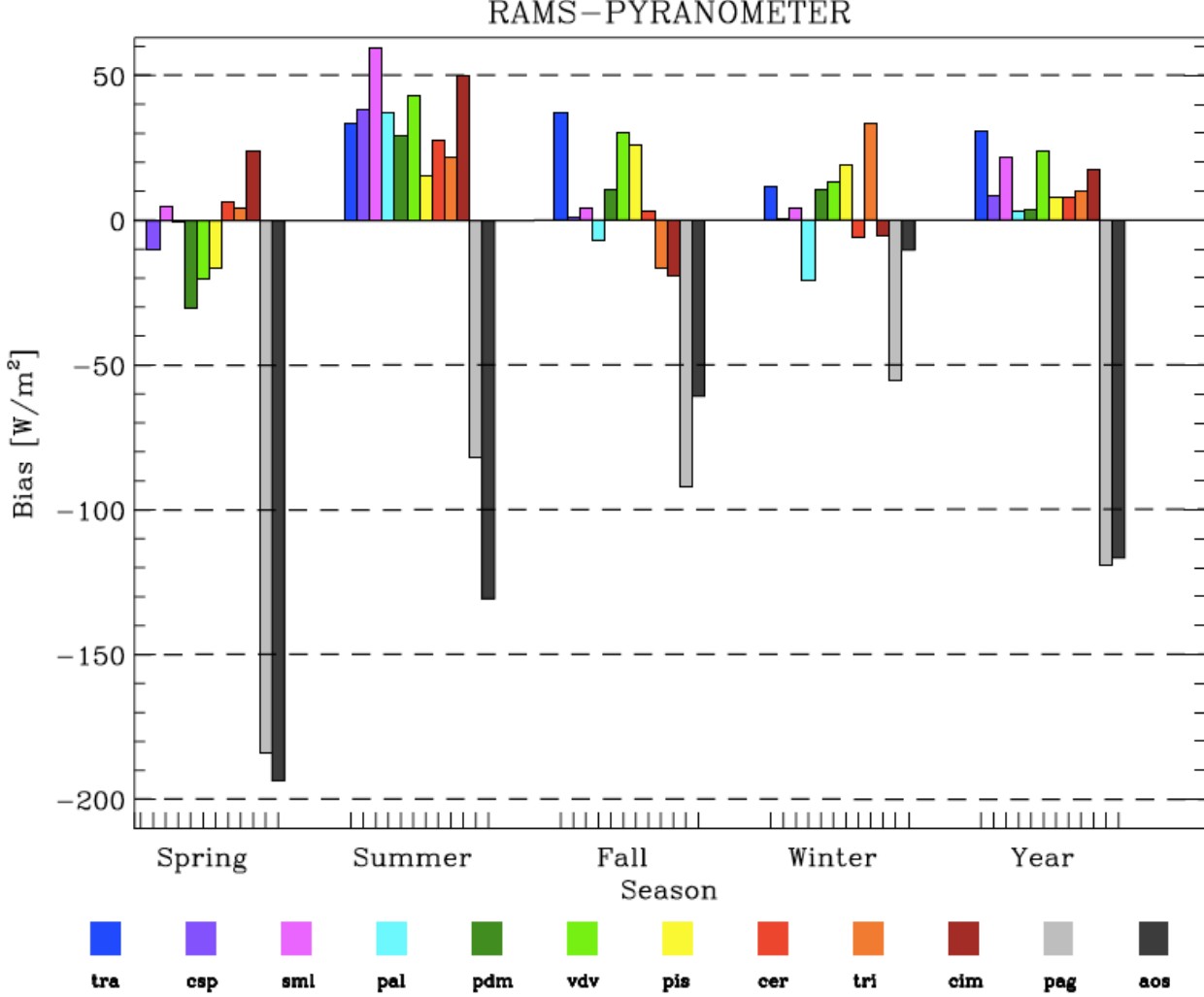


Figure 5: a) MBE for the MSG-GHI for the different stations and seasons as well as for the whole
year. The figure has been derived from the hourly data of pyranometers and MSG-GHI estimate; b)
As in Figure 5a for the RAMS forecast.









**a)**

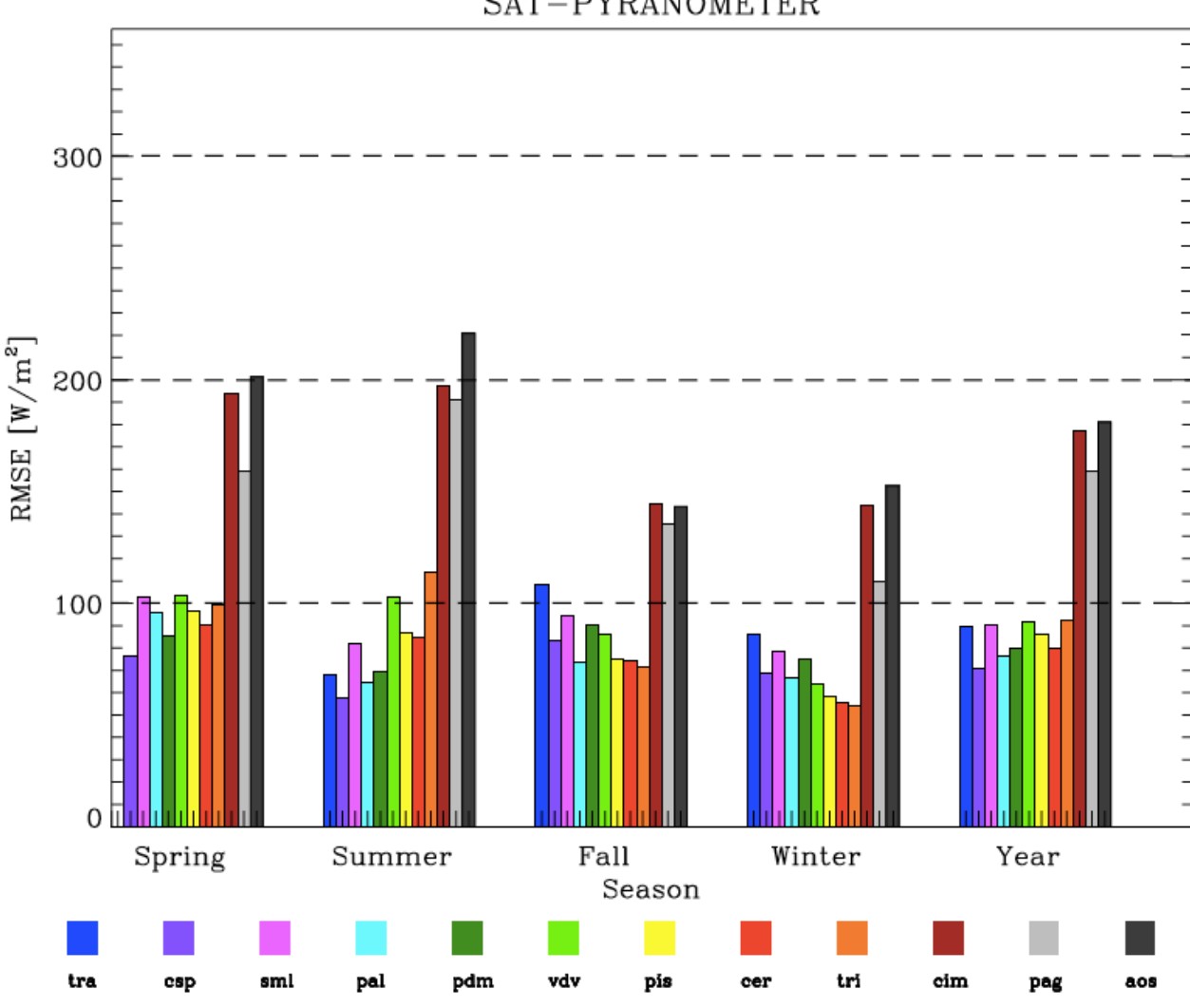

**b)**

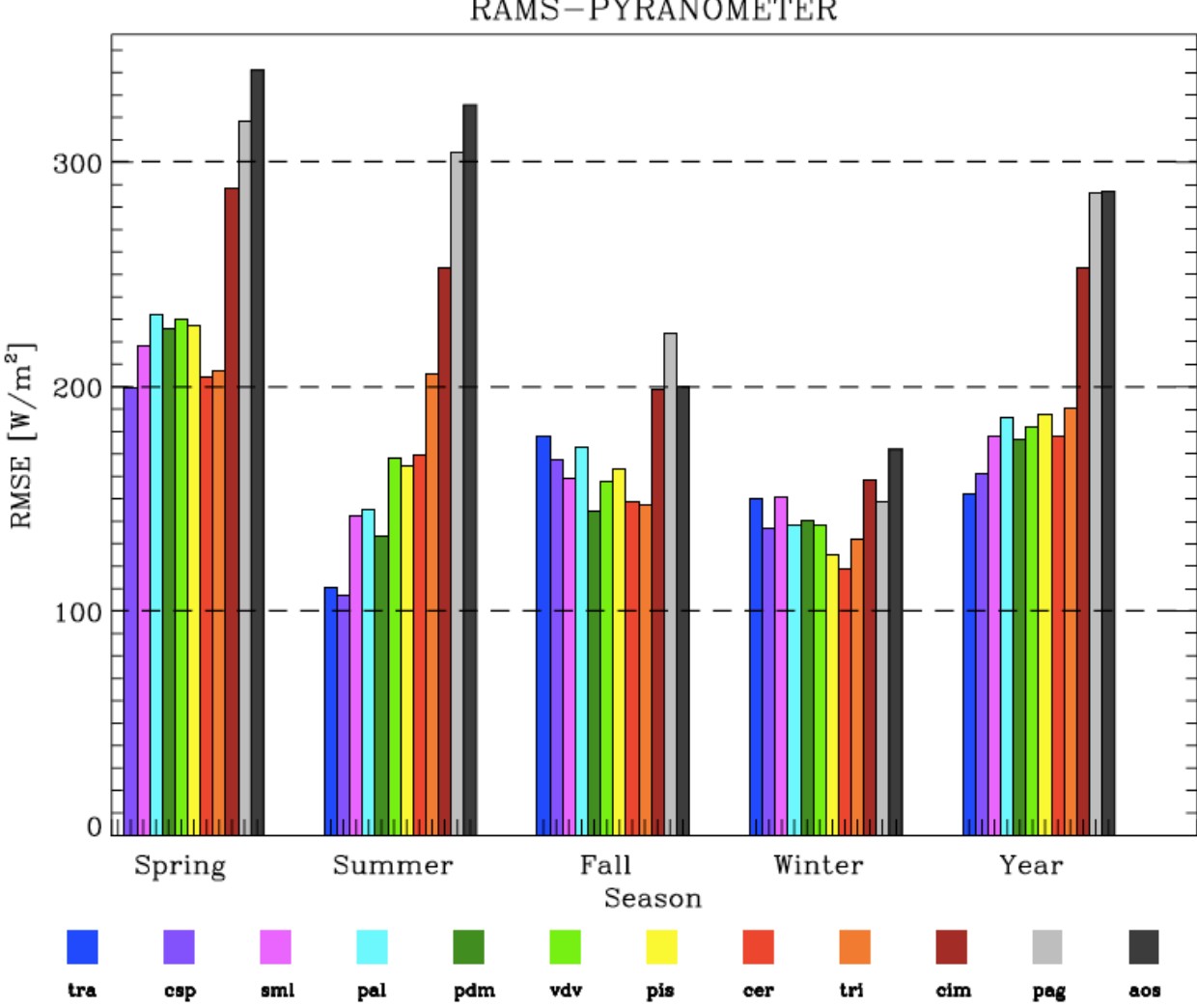


Figure 6: a) RMSE for the MSG-GHI for the different stations and seasons as well as for the whole
year. The figure has been derived from the hourly data of pyranometers and MSG-GHI estimate; b)
As in a) for the RAMS forecast.








**a)**

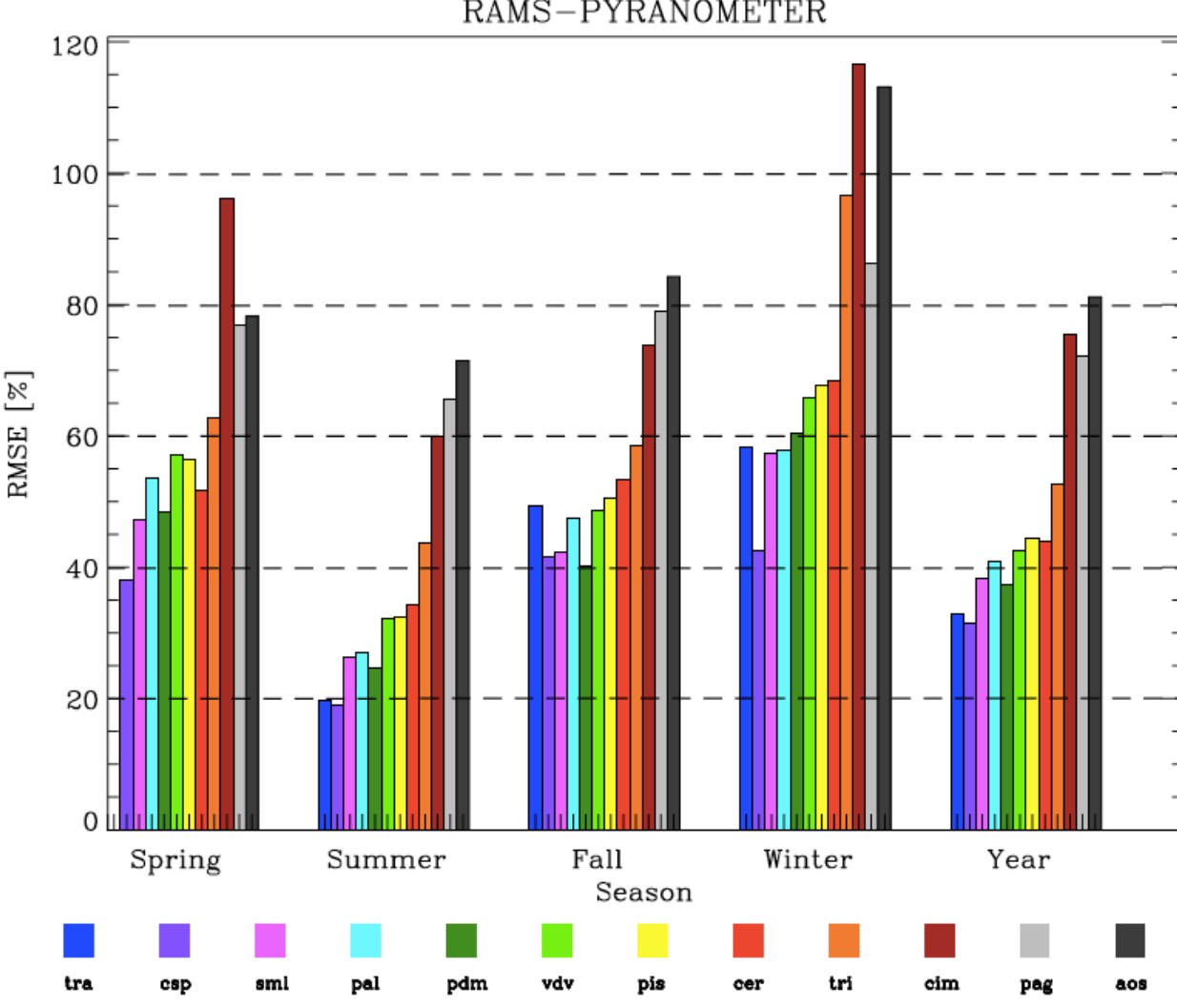


**b)**

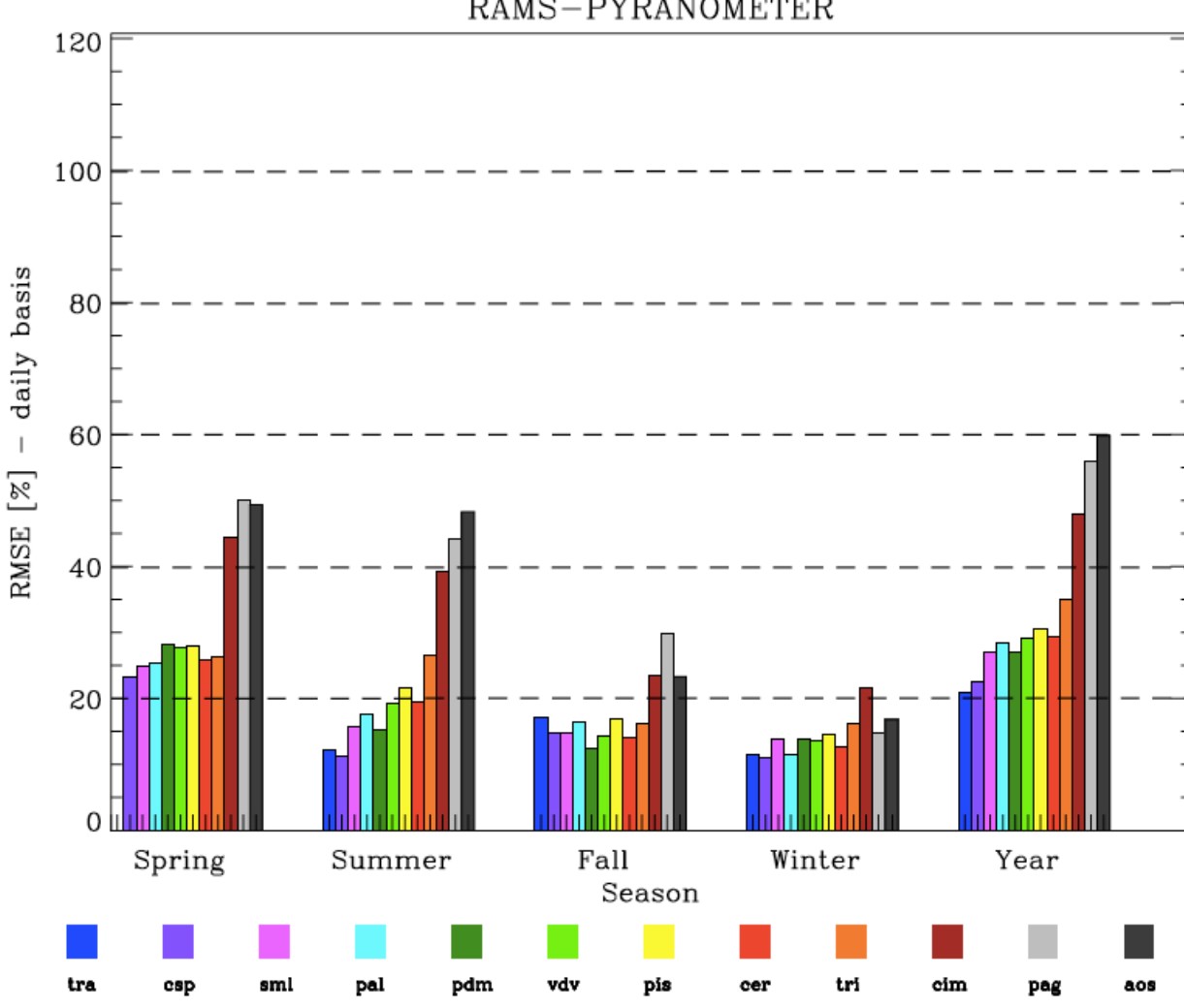


**c)**

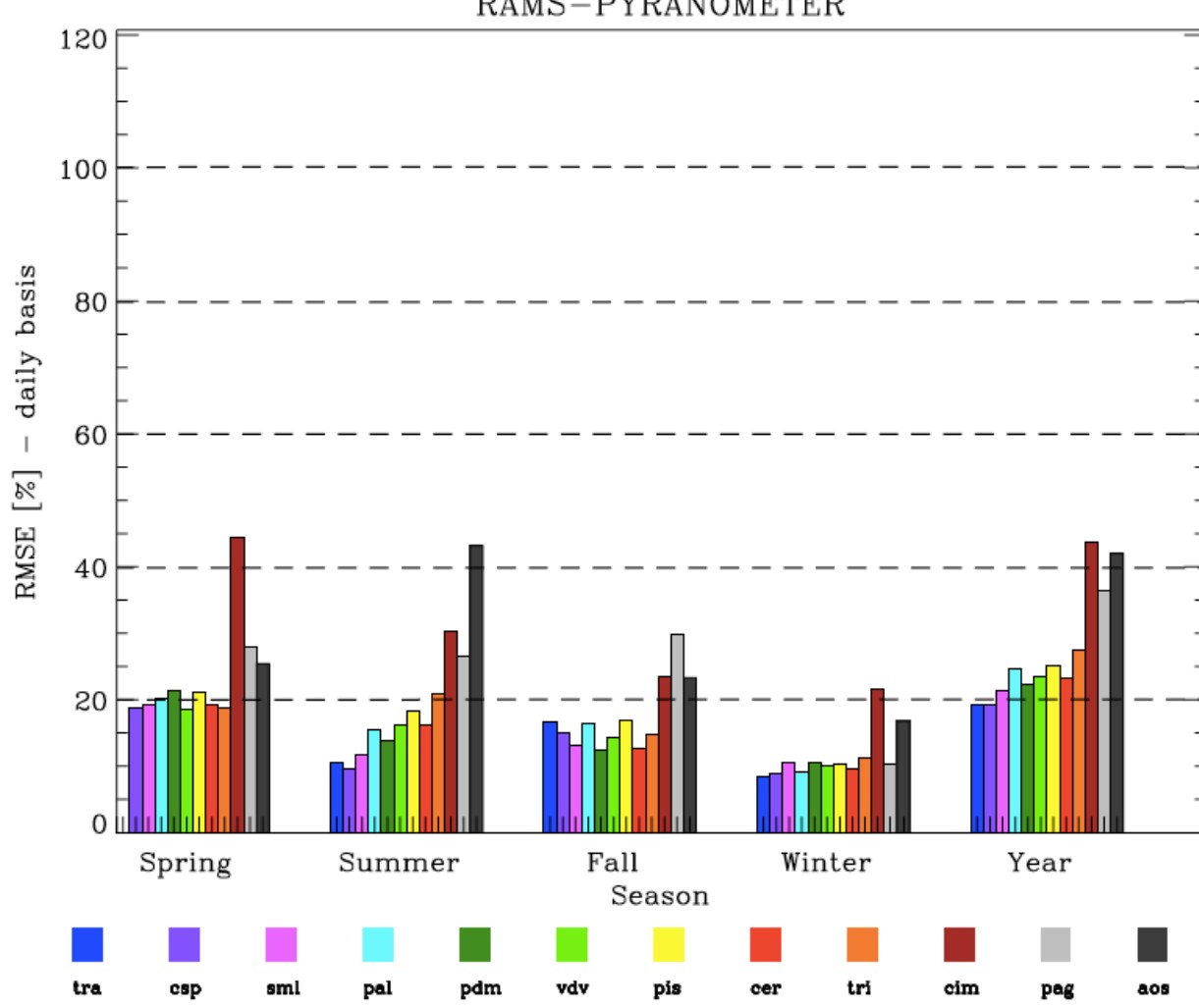



Figure 7: a) rRMSE computed for different seasons and stations, as well as for the whole year, for
the RAMS-GHI one-day hourly forecast; b) as in a) for daily integrated GHI; c) as in b) after the
MOS correction to the model output.