# Peer review of "Comparison of hourly surface downwelling solar radiation estimated from MSG/SEVIRI and"

_Atmospheric Measurement Techniques, 2017_

## Referee Comment (RC2)

**REVIEW OF MANUSCRIPT SUBMITTED TO Atmos. Meas. Tech. Discuss.**

Authors: Stefano Federico, Rosa Claudia Torcasio, Paolo Sanò, Daniele Casella, Monica Campanelli, Jan Fokke Meirink, Ping Wang, Stefania Vergari, Henri Diémoz, Stefano Dietrich

Title: Comparison of hourly surface downwelling solar radiation estimated from MSG/SEVIRI and forecast by RAMS model with pyranometers over Italy

Date of review: 02 APR 2017

OVERALL EVALUATION

The manuscript presents a one-year comparison between satellite-estimated and numerical weather prediction model –forecasted solar radiation with ground-based measurements at Italian sites. The paper is of interest as it involves both weather forecasting and satellite algorithms, considering a topical subject connected to solar energy production.

The manuscript presents a fairly thorough evaluation of the performance of these two sources of solar radiation information for the chosen Italian sites. The figures presented and the used equations seem correct. However, the manuscript occasionally presents conclusions that are not supported by the evidence presented in this study, and furthermore suffers from unclear sentences that perhaps could be improved through a proper language check / proof reading.

I would recommend major revisions before accepting this manuscript. Note, however, that I do not believe that the actual scientific work will require a deep revision, but rather, that the authors need to pay attention to way things are expressed and what conclusions can be made based on the results presented in their manuscript.

SOMEWHAT GENERAL COMMENTS

- L59-61 and elsewhere: throughout the manuscript, it would be important to emphasize (and remind the reader of the fact) that RAMS is a forecast (for the day ahead), and MSG-GHI is a satellite-based estimate (available some time after the satellite observations have been made). This needs to always be kept in mind when comparing the performance of the two – the present manuscript is occasionally somewhat sloppy on this.
- L110—118: The RAMS model should be properly introduced before starting the paragraph on exchange between atmosphere and surface. What is the RAMS model?
- L151—164: The text seem somewhat unclear here, should be clarified. It seems to me that maybe there are two different groups of pyranometers used: (i) L151-157, and (ii) L158-164. The stability of the pyranometer in

Aosta is documented, but nothing is said about the other pyranometers. If sunshine duration is used (point 2), which stations have sunshine duration available? How is the Aosta check against libRadtran done, what are the criteria for data removal? Which institutes are responsible for the pyranometer measurements?

- Fig 6 + analysis: The manuscript up to this point has explained/speculated about the role of clouds in the performance. Now, Fig 6 finally shows quantitative results on how the performance behaves as a function of cloud classification. To me, it would make sense to bring this figure more toward the early part of the manuscript, so that the remaining analysis already could make use of these results as this would reduce the need for indirect determination.
- Conclusions -> (suggestion) Summary and Conclusion: The section is in its present form to a large extent repeating/summarizing the results presented in previous sections. It would be more interesting for the reader if some more discussion/conclusions would be added. Perhaps the section could also be shortened.
- L448: Are any evidence presented in the manuscript that show that the radiative scheme is inable to simulate cloudy conditions correctly? Where does this statement come from?

SPECIFIC SUGGESTIONS

- L25-26: this is not always true (see e.g. L445—446)
- L27: RMSE increases for Alpine stations (and similar statements elsewhere). This seems a bit misleading. I would suggest saying "is higher" or "is lower".
- L30: "RMSE ranges from 152 W/m2 to…" – here (and elsewhere) it should be defined that the RMSE has been calculated for hourly values.
- L36: "a reduction" -> "lower"
- L36: "of at least 10%" – here, it needs to defined what 10% means (10% of the base value, or a change corresponding to 10% steps/units (e.g. from 20 to 10%). Also elsewhere.
- L46: two specific papers are cited for a scope very wide. I would suggest removing the references.
- L51: Yes, PV can convert GHI to electricity, but much more commonly, they convert tilted GHI, or perhaps more generally, just solar radiation, to electricity
- L91: suggest to remove "So"
- L92: particle size is not an optical property
- L94: could you clarify the text here, is a mixture of ice/water clouds possible?
- L112: "most of Europe" seems a bit exaggerated
- L131-132: Somewhat unclear what exactly this means. Could be elaborated more.
- L133-136: Please clarify, are there any additional data assimilated into the RAMS model, e.g. weather observations, or is the RAMS model's initial state fully determined purely by ECMWF?

- L138-139: Seems contradicting that a weather forecasting model would need a spin up time of 12 hours.
- Section 2.3: could be separated into two: (i) surface observations / (ii) evaluation methodology
- L146: "Vigna di Valle is still" -> "Vigna de Valle is" (remove still)
- L165: "environmental characteristics" seem to actually mean cloud classification by the satellite method, is that correct? Please clarify text and use suitable terminology.
- L170: (language) "with the stations" -> "between different stations" ?
- L174-175: somewhat unclear sentence, please clarify
- L178-179 + L188-189: why not use equation numbers?
- Figure 3: I would suggest swapping the axes, so that pyranometer values are on the x-axis and estimated values on the y-axis. This makes values above the 1:1 line correspond to overestimation and vice versa, which is more logical. Also, I think it would be interesting to add this kind of figure for each station as a supplement or appendix as some readers will be interested in that information. Finally, the figure would be easier to read if grid lines and a legend were added, and if the point style would be modified so that points would not overlap (as much) in the busy areas of the plot.
- L196 / Fig. 3: clarify how the regression lines were determined
- L201-202 and L220: "it is apparent the larger scatter" -> (language) please rephrase (also similar sentence construction elsewhere)
- L229-230: in point b, it needs to be emphasized that RAMS is a forecast and thus not directly comparable to MSG
- L256-257: It is unclear to the reader how the conclusion about clouds being the main source of error was made. Could this be elaborated?
- L257-264: This seems to be mostly somewhat loose speculation, although things are expressed as hard facts. The evidence presented in the manuscript does not support all these statements. Therefore, I recommend rewriting, to use more careful statements.
- L268-274: On a general level, I believe the explanations presented here to be plausible. However, I also find that the authors focus too much on explanations that have to do with local orography and horizontal resolution. Could there be something else involved as well? For example, one factor that certainly plays a role here is the fact that mountain stations have more clouds and clouds are difficult for the satellite algorithm (as seen later on in Fig 6).
- L287: RMSE -> rRMSE?
- L288-289: unclear to me what is meant by "statistic shows more clearly the impact of ..."
- L294 + Fig 5 caption: "RAMS-GHI one-day forecast". Here (and elsewhere when mentioning RAMS one-day forecast) it would be important to emphasize that RMSE is based on hourly values of the day ahead forecast from RAMS. The present text leads me to think that values may be daily.
- L301: I find it odd to say "This result is caused by RMSE statistics".
- L321: "all sky conditions, which showed" -> (suggest) "all sky conditions, which indirectly showed..."
- L329-331: Unclear how this explains the difference?

- L333: Not completely true, compare with L445-446
- L344-346: Please clarify how the persistence forecast is created. Are values hour by hour assumed to be the same between the two days?
- L345: I find the use of the short-version "1D" somewhat misleading (as it makes me think of one-dimensional) and therefore suggest writing it out: one-day.
- Section 3.3: split into two separate sections? (i) Daily evaluation / (ii) MOS application
- L381-383: There may also be other sources of MBE
- L384: "The MOS consists of" -> (suggestion) "The MOS used here consists of"
- L389-393: Unclear how exactly this works, please clarify.
- L402: Somewhat unclear how this conclusion can be made based on the above sentences
- L436-440 / L441-446: the latter paragraph presents discussion on performance as function of cloud classification, while the previous contains similar information, but indirectly. Maybe the paragraphs could be combined into one, or the order be changed, to make more effective and convincing communication.

---

## Short Comment (SC1) · 15 Mar 2017

Dear authors,

I wanted to offer a few comments and suggestions regarding the RAMS model that may offer improvements to your forecasts.

1. There is a much newer and supported version of RAMS maintained at Colorado State University (CSU-RAMS) that is freely available at the following URL:

http://vandenheever.atmos.colostate.edu/vdhpage/rams.php

2. This version is based on RAMS-6 and is consistently improved and updated.

3. This version has the Harrington (1997) aerosol- and hydrometeor-sensitive radiation scheme. This radiation representation would be much more realistic since it accounts for radiative impacts of each liquid and ice hydrometeor species.

4. The RAMS microphysics is an excellent and well-proven microphysics scheme. The microphysics in the latest CSU-RAMS has undergone great improvements in recent years. A list of model development references can be found at:

http://vandenheever.atmos.colostate.edu/vdhpage/rams/rams_dev_pubs.php

And an updated list of RAMS scientific usage references can be found at:

http://vandenheever.atmos.colostate.edu/vdhpage/rams/rams_use_pubs.php

Using a two-moment microphysics package for studying radiative effective of clouds may have been a better choice than using the interfaced WRF single moment scheme. The single moment scheme doesn't account for changes in hydrometeor number concentration which can be important for radiation scattering and absorption.

See the following paper discussing 1 vs 2 moment microphysics schemes:

Igel, A.L., M.R. Igel, and S.C. van den Heever, 2015: Make it a double? Sobering results from simulations using single-moment microphysics schemes. J. Atmos. Sci., 72, 910-925.

I think many of the model deficiencies in your study could be linked to use of the simplified Chen-Cotton radiation, Kuo convective parameterization, and WRF single-moment micro.

I would suggest usage of Harrington radiation, Kain-Fritsch cumulus parameterization (where appropriate), and RAMS 2-moment micro. Each of these is available in the CSU RAMS version.

Perhaps a comparison between your version of RAMS and the latest CSU-RAMS would offer insight into both versions of RAMS and assist in model improvement in either version and potentially lead to improved forecasts of radiative quantities.

---

## Referee Comment (RC1) · Anonymous Referee #1 · 21 Mar 2017

This study uses ground-based measurements in Italy as to evaluate the GHI estimations from MSG and RAMS in hourly basis. Then, a correction technique was performed in order to improve the comparison results.

The scientific quality and presentation of the paper is very good and with some minor technical corrections and revisions, it could be a valuable publication in the AMT journal.

First of all the MOS correction needs to briefly described in the "Data and Methods" section in order to provide a connection with the application results in sub-section 3.3.

[Figure]

The authors could provide some references for the uncertainties in cloud-properties and MACC outputs, and discuss the sensitivity of these parameters to the MSG-GHI evaluated values.

The same literature-based uncertainty and sensitivity analysis need to be discussed for the RAMS, as to directly connect with the evaluation results presented, by providing comparable specific ranges and values as well.

Finally, the conclusions section need to be merged into some additional general findings, highlighting the innovation and value of this study.

Overall, the presented techniques are scientifically sufficient, the results are well determined and falls into the scope of AMT, so I believe that after the above minor corrections, the paper can be published.

---

## Author Comment (AC2) · 12 Apr 2017

We acknowledge Stephen Saleeby for the precise comment on the RAMS (Regional Atmospheric Modeling System) used in this paper. Our answer is detailed below.

The motivation to use this specific version of the RAMS model is that this model is operational at ISAC-CNR and we aimed to test this specific version. This model uses the WSM6 microphysical scheme that was interfaced with RAMS at ISAC-CNR (Federico, 2016) and that has good performance over Italy. Other parameterizations were included in the RAMS model at ISAC-CNR, as the lightning scheme (Federico et al., 2014); these parameterizations, however, do not play a role for the present study.

Of course, we acknowledge the important developments that were done at the CSU (CSU-RAMS), and that should improve the model performance for the GHI forecast. The comparison between the RAMS used in this paper and the CSU-RAMS will be performed soon, to support quantitatively the difference (and the expected improvement) among the two models.

For this paper, following also the comment of the first reviewer, we added a comment on the expected improvement using the latest development included in CSU-RAMS. We will write in the "Summary and Conclusions" section:

[revised manuscript text omitted]

---

## Author Comment (AC3) · 13 Apr 2017

We acknowledge the reviewer for the precise and useful comments.

Please also note the supplement to this comment:
http://www.atmos-meas-tech-discuss.net/amt-2017-9/amt-2017-9-AC3-supplement.pdf

---

## Author Comment (AC1)

We acknowledge the reviewer for the useful comments that will improve the paper. In the following, there are our answer to each comment.

**Reviewer #1**

This study uses ground-based measurements in Italy as to evaluate the GHI estimations from MSG and RAMS in hourly basis. Then, a correction technique was performed in order to improve the comparison results.
The scientific quality and presentation of the paper is very good and with some minor technical corrections and revisions, it could be a valuable publication in the AMT journal.
First of all the MOS correction needs to briefly described in the "Data and Methods" section in order to provide a connection with the application results in sub-section 3.3.

-We will move the MOS description in the "Data and Methods" section.

The authors could provide some references for the uncertainties in cloud-properties and MACC outputs, and discuss the sensitivity of these parameters to the MSG-GHI evaluated values.

We will comment on the uncertainty in cloud properties and MACC output, adding some references on the subject. We will write:

"The retrieval of cloud properties can be associated with large uncertainties, in particular due to horizontal inhomogeneity (e.g., Coakley et al., 2005). However, subsequently derived irradiances (such as SICCS GHI) have relatively much smaller uncertainty due to compensation of errors in forward and inverse radiative transfer calculations (Greuell et al., 2013; see also Kato et al., 2006). Uncertainties in MACC reanalysed aerosol properties contribute to errors in retrieved clear-sky GHI but these errors are considerably smaller than those for cloudy skies (Greuell et al., 2013)."

The same literature-based uncertainty and sensitivity analysis need to be discussed for the RAMS, as to directly connect with the evaluation results presented, by providing comparable specific ranges and values as well.

We acknowledge the reviewer and Stephen Saleeby for this specific comment on the RAMS model. We will clarify the uncertainties that are expected with our version of the RAMS model, considering recent developments to the model as well as results obtained with different models.

[revised manuscript text omitted]

Finally, the conclusions section need to be merged into some additional general findings, highlighting the innovation and value of this study.

Considering this comment and that of the second reviewer about the conclusions, the "Conclusion" section ("Summary and Conclusions" in the revised version) will be shortened about the results of this paper, while the results of the paper will be compared with similar studies in other Mediterranean countries. In general, the results of this paper span a wider performance compared to other studies, because of the very different climatic characteristics of the stations. Also, the above discussion on the RAMS model will be included in the "Summary and Conclusions" section.

Overall, the presented techniques are scientifically sufficient, the results are well determined and falls into the scope of AMT, so I believe that after the above minor corrections, the paper can be published.

---

## Author Response (AR1)

Dear Editor,

First of all, we acknowledge both Reviewers and Stephen Saleeby for their useful comments and for the precise review that improved the quality of the paper. All comments were considered and our answers are detailed below in blue.

**Reviewer #1**

This study uses ground-based measurements in Italy as to evaluate the GHI estimations from MSG and RAMS in hourly basis. Then, a correction technique was performed in order to improve the comparison results.
The scientific quality and presentation of the paper is very good and with some minor technical corrections and revisions, it could be a valuable publication in the AMT journal.
First of all the MOS correction needs to briefly described in the "Data and Methods" section in order to provide a connection with the application results in sub-section 3.3.

We have moved the MOS description in the "Data and Methods" section.

The authors could provide some references for the uncertainties in cloud-properties and MACC outputs, and discuss the sensitivity of these parameters to the MSG-GHI evaluated values.

We have commented a bit on the uncertainty in cloud properties and MACC output, adding some references on the subject. We wrote:

"The retrieval of cloud properties can be associated with large uncertainties, in particular due to horizontal inhomogeneity (e.g., Coakley et al., 2005). However, subsequently derived irradiances (such as SICCS GHI) have relatively much smaller uncertainty due to compensation of errors in forward and inverse radiative transfer calculations (Greuell et al., 2013; see also Kato et al., 2006). Uncertainties in MACC reanalysed aerosol properties contribute to errors in retrieved clear-sky GHI but these errors are considerably smaller than those for cloudy skies (Greuell et al., 2013)."

The same literature-based uncertainty and sensitivity analysis need to be discussed for the RAMS, as to directly connect with the evaluation results presented, by providing comparable specific ranges and values as well.

We acknowledge the Reviewer and Stephen Saleeby for this specific comment on the RAMS model. We clarified the uncertainties that are expected with our version of the RAMS model, considering recent developments to the model as well as results obtained with different models. We wrote:

[revised manuscript text omitted]

Finally, the conclusions section need to be merged into some additional general findings, highlighting the innovation and value of this study.

Considering this comment and that of the Reviewer #2 about the conclusions, the "Conclusion" section ("Summary and Conclusions" in the revised version) has been shortened about the statistics shown in the paper, while the results have been compared with similar studies in other Mediterranean countries (Greece and Spain, see the answer to the Reviewer #2 for details). Also, a discussion on the specific version of the RAMS model used in this paper (above comment) has been included considering the recent developments of the model.

Overall, the presented techniques are scientifically sufficient, the results are well determined and falls into the scope of AMT, so I believe that after the above minor corrections, the paper can be published.

**Reviewer #2**

**REVIEW OF MANUSCRIPT SUBMITTED TO Atmos. Meas. Tech. Discuss.**

Authors: Stefano Federico, Rosa Claudia Torcasio, Paolo Sanò, Daniele Casella, Monica Campanelli, Jan Fokke Meirink, Ping Wang, Stefania Vergari, Henri Diémoz, Stefano Dietrich

Title: Comparison of hourly surface downwelling solar radiation estimated from MSG/SEVIRI and forecast by RAMS model with pyranometers over Italy

Date of review: 02 APR 2017

OVERALL EVALUATION

The manuscript presents a one-year comparison between satellite-estimated and numerical weather prediction model –forecasted solar radiation with ground- based measurements at Italian sites. The paper is of interest as it involves both weather forecasting and satellite algorithms, considering a topical subject connected to solar energy production.

The manuscript presents a fairly thorough evaluation of the performance of these two sources of solar radiation information for the chosen Italian sites. The figures presented and the used equations seem correct. However, the manuscript occasionally presents conclusions that are not supported by the evidence presented in this study, and furthermore suffers from unclear sentences that perhaps could be improved through a proper language check / proof reading.

I would recommend major revisions before accepting this manuscript. Note, however, that I do not believe that the actual scientific work will require a deep revision, but rather, that the authors need to pay attention to way things are expressed and what conclusions can be made based on the results presented in their manuscript.

SOMEWHAT GENERAL COMMENTS

- L59-61 and elsewhere: throughout the manuscript, it would be important to emphasize (and remind the reader of the fact) that RAMS is a forecast (for the day ahead), and MSG-GHI is a satellite-based estimate (available some time after the satellite observations have been made). This needs to always be kept in mind when comparing the performance of the two – the present manuscript is occasionally somewhat sloppy on this.

-Ok. Clarified throughout the paper. Where appropriate we used: "RAMS-GHI one-day hourly forecast"

- L110—118: The RAMS model should be properly introduced before starting the paragraph on exchange between atmosphere and surface. What is the RAMS model?

- Thank you for noting this point. We wrote: "RAMS is a general purpose limited area model designed to be used at the mesoscale (horizontal grid spacing ≈ 1-100 km) or higher horizontal resolutions. It is based on a full set of non-hydrostatic, compressible equations of the atmospheric dynamics and thermodynamics, plus conservation equations for scalar quantities such as water vapour and liquid and ice hydrometeor mixing ratios. The model is widely used for research as well as for weather forecast (Cotton et al., 2003)."

- L151—164: The text seems somewhat unclear here, should be clarified. It seems to me that maybe there are two different groups of pyranometers used: (i) L151-157, and (ii) L158-164. The stability of the pyranometer in Aosta is documented, but nothing is said about the other pyranometers. If sunshine duration is used (point 2), which stations have sunshine duration available? How is the Aosta check against libRadtran done, what are the criteria for data removal? Which institutes are responsible for the pyranometer measurements?

The pyranometers are managed by two different institutions and each institution is responsible for its own observations. The Aosta pyranometer is managed by Arpa Valle D'Aosta, while all other pyranometers are managed by the Aeronautica Militare. The check with the LibRadtran software for Aosta is made to test for electric wiring faults. In particular, measurements higher than 200% of the daily maximum expected from libRadtran in clear-sky conditions are removed. A comment was added for the stability of the Italian Air Force (Aeronautica Militare) pyranometers.

To clarify these points, we wrote:

"The pyranometers are managed by two different institutions. The Aosta pyranometer is managed by Arpa Valle D'Aosta, while all other pyranometers are managed by the Italian Air Force (Aeronautica Militare). Each institution is responsible for its own measurements.

For pyranometers managed by the Italian Air Force, in addition to basic maintenance and installing procedures recommended by WMO – Guide nr. 8, data quality is controlled following an internal control procedure described in Vergari et al. (2010).

In particular, to improve quality control checks for global solar radiation and sunshine duration data (available simultaneously for all stations of this paper managed by Aeronautica Militare), two procedures have been implemented. A range limit check, applied to both variables separately, concerns the respect of variables' physical limits. This check has been improved varying physical limits in agreement to the latitude and the season. Furthermore, the monthly atmospheric clearness index has been calculated from the climatic history of each site, by applying the linear form of the Angstrom-Prescott model. Then, an upper and a lower bound for the solar radiation are defined as linear functions of clearness index and the sunshine duration value. These bounds delimit the range of the daily solar radiation.

Analyzing the distance of daily values from their bounds, it is also possible to prevent instrumental electronic drifts. In fact, if this distance changes in an appreciable way, a recalibration procedure is activated and the device is recalibrated by comparison with a standard pyranometer using the sun as a source, under natural conditions of exposure (ISO ,1993). The reference standard used in this case is a CM11 Kipp and Zonen, calibrated every two year by the WMO Regional Instrument Centre Radiation of Carprentrass (France), by comparison with a pyreliometer PMO6 and a pyranometer CMP21.

For the Aosta pyranometer, in addition to the manual maintenance related to the periodical cleaning of the dome, irradiance measurements are daily checked through comparison with clear-sky simulations by a radiative transfer model (libRadtran, Emde et al., 2016) to check for electric wiring faults. In particular, measurements higher than 200% of the daily maximum expected from libRadtran in clear-sky conditions are removed. The CMP21 radiometer is calibrated every two years at the Physikalisch-Meteorologisches Observatorium Davos/World Radiation Center (PMOD/WRC) against a member of the World Standard Group (WSG) for the direct component and a shaded standard pyranometer of the World Radiation Center (WRC) for the diffuse component. The radiometric stability was better than 0.2% over the period of the six years of measurements."

- Fig 6 + analysis: The manuscript up to this point has explained/speculated about the role of clouds in the performance. Now, Fig 6 finally shows quantitative results on how the performance behaves as a function of cloud classification. To me, it would make sense to bring this figure more toward the early part of the manuscript, so that the remaining analysis already could make use of these results as this would reduce the need for indirect determination.

- Thank you for noting this point. We moved the analysis of Figure 6 (Figure 4 in the revised version) at

the end of Section 3.1 to show the impact of the cloud coverage on the RAMS-GHI forecast and MSG-GHI estimation performance for all stations. We believe that this clarifies the following analysis, as you suggested.

  - Conclusions -> (suggestion) Summary and Conclusion: The section is in its present form to a large extent repeating/summarizing the results presented in previous sections. It would be more interesting for the reader if some more discussion/conclusions would be added. Perhaps the section could also be shortened.

-Considering this comment and that of the Reviewer #1 about the conclusions, the "Conclusion" section ("Summary and Conclusions" in the revised version) has been shortened about the results of this paper, while the results of the paper have been compared with similar studies in other Mediterranean countries. Also, a discussion on the specific version of the RAMS model used in this paper will be included considering the recent developments to the model. The discussion on the specific version of the RAMS model is reported in the answer to the Reviewer #1, while here we show the discussion on the comparison with similar studies in the Mediterranean countries (Spain, Greece). We wrote:

"To put the results of this paper in a more general context, we compare our statistics with similar studies in the Mediterranean area (Greece and Spain).

Kosmopulos et al. (2015) quantified the performance of the MM5 model for the one- and two-days forecast over Greece. The forecast was compared with eleven pyranometers displaced over the country. The RMSE computed from hourly data and for the one-day forecast ranges between 160 W/m$^2$ for the Chania station to 230 W/m$^2$ for Amfiklia. The error increases with the terrain complexity and cloud coverage: Chania is located in the western part of the Crete Island and shows a Mediterranean climate, while Amfiklia is located in one of the highest plateaus of Greece, bounded at the west by the Pindos mountain. The RMSE shows a small increase between the first and second day of forecast. With the exception of the mountainous stations of this paper, where the RMSE is larger, our performance is in line with that of Kosmopulos et al. (2015). Also, both studies show a positive MBE with values of few tens of W/m$^2$ for most stations, with the exception of Paganella and Aosta stations of this study where the MBE is larger in absolute value.

Gómez et al. (2016) quantified the performance of the RAMS model (both versions 4.4 and 6.0) for the one-, two- and three-days GHI forecast over the Valencia Region. They considered thirteen pyranometers widespread over the region. Focusing on the RMSE for hourly data in summer, they found errors of 200 W/m$^2$ for flat terrain and 250 W/m$^2$ for hilly terrain. The RMSE for winter is 150-160 W/m$^2$, depending on the stations. The MBE is of few tens of W/m$^2$ and it is positive. They found similar results among the three days of forecast and also between the two versions of the RAMS model. With the exceptions of the mountainous stations of this paper, where both the RMSE and MBE in absolute value are larger, our results are in line with those of Gómez et al. (2016).

Lara Fanego et al. (2012) examined the performance of the WRF model for the GHI one- two- and three-days forecast over Andalucia (Spain). They consider four stations: Andasol, Jerez, Cordoba and Huelva. The RMSE computed from hourly data for the whole year is 140 W/m$^2$ for Cordoba, Jerez ad Huelva stations and 170 W/m$^2$ for Andasol. Differences of the RMSE among the three days of forecast are small. The RMSE of Lara Fanego et al. (2012) is smaller (10-20 W/m$^2$) than those of this paper.

This result can be caused by the difference of the climate and orography at the stations considered in the two studies, nevertheless a better treatment of the interaction between aerosols and radiation in Lara Fanego et al. (2012) contribute to this difference. The MBE of Lara Fanego et al (2012) is in line with that of this paper, with the exception of Paganella and Aosta stations."

- L448: Are any evidence presented in the manuscript that show that the radiative scheme is unable to simulate cloudy conditions correctly? Where does this statement come from?

- We agree that the worse simulation of the GHI in cloudy conditions in not necessarily a consequence of the radiative scheme as other errors, for example the estimation of the hydrometeor concentrations, have a role. We modified the sentence and we wrote: "The increase of the RMSE with the cloud coverage is a combination of both the inability of the two methods to correctly represent the cloud coverage and of the difficulty to compute the GHI in cloudy conditions."

SPECIFIC SUGGESTIONS

- L25-26: this is not always true (see e.g. L445—446).

We modified the sentence: "Results for hourly data show an evident dependence on the sky conditions, with the Root Mean Square Error (RMSE) increasing from clear to cloudy conditions."

- L27: RMSE increases for Alpine stations (and similar statements elsewhere). This seems a bit misleading. I would suggest saying "is higher" or "is lower".

- Corrected according to the Reviewer comment.

- L30: "RMSE ranges from 152 W/m2 to..." – here (and elsewhere) it should be defined that the RMSE has been calculated for hourly values.

- Corrected according to the Reviewer comment.

- L36: "a reduction" -> "lower"

-Ok. We wrote "Results for daily integrated GHI show lower RMSE compared to hourly GHI evaluation for both RAMS-GHI one-day forecast and MSG-GHI estimate. Considering the yearly evaluation, the RMSE of daily integrated GHI is at least 9% lower (in percentage units, from 31% to 22% for RAMS for Cozzo Spadaro) than the RMSE computed for hourly data for each station."

- L36: "of at least 10%" – here, it needs to defined what 10% means (10% of the base value, or a change

corresponding to 10% steps/units (e.g. from 20 to 10%). Also elsewhere.

-Thanks for this point: the value refers to a change in the percentage units (as in the previous comment). Clarified throughout the paper.

- L46: two specific papers are cited for a scope very wide. I would suggest removing the references.

-Done.

- L51: Yes, PV can convert GHI to electricity, but much more commonly, they convert tilted GHI, or perhaps more generally, just solar radiation, to electricity.

- Corrected. We used "solar radiation".

- L91: suggest to remove "So"

- Done.

- L92: particle size is not an optical property

- Deleted.

- L94: could you clarify the text here, is a mixture of ice/water clouds possible?

- Clarified: a mixture of ice/water cloud is not possible.

- L112: "most of Europe" seems a bit exaggerated

- Corrected: "Central Europe"

- L131-132: Somewhat unclear what exactly this means. Could be elaborated more.

We added the following part to clarify the point: "In particular, the scheme uses an "effective emissivity" for cloud layers, where the cloud emissivity is parametrized empirically from observations (Stephens 1978). The "effective emissivity" is a function of the total condensate water path, computed summing all hydrometeors mixing ratios (liquid, i.e. cloud and rain, solid, i.e. ice and snow, and mixed phase, i. e. graupel) and integrating over the cloud-layer (Chen and Cotton, 1983).". In the "Summary and Conclusion" section it is also discussed the important impact of taking into account for the different phases of the condensate water.

- L133-136: Please clarify, are there any additional data assimilated into the RAMS model, e.g. weather observations, or is the RAMS model's initial state fully determined purely by ECMWF?

No additional data are assimilated into the RAMS model and the initial state is fully determined by the ECMWF. We wrote: "No additional data are assimilated into the RAMS model."

- L138-139: Seems contradicting that a weather forecasting model would need a spin up time of 12 hours.

- The model configuration of this paper uses a cold start with no hydrometeors, with the exception of the water vapour, at initial time. Previous unpublished studies with RAMS showed that 12 h are enough for the model to reach a dynamical equilibrium between the dynamic, thermodynamic and cloud-precipitation fields starting from a cold start. The 6 h spin-up time is enough for most cases, but there are occasions where a longer spin-up time is required. We consider this point as follows:

"The model was run for a whole year (1 June 2013 - 31 May 2014) with the above configuration and with no hydrometeors at the initial time, with the exception of water vapour (cold start). Previous unpublished studies with RAMS showed that 12 h are enough for the model to reach a dynamical equilibrium between the dynamic, thermodynamic and cloud-precipitation fields starting from a cold start. For this reason, each simulation lasts 36 h, starts at 12 UTC of the day before the day of interest, and the first 12 h are used as spin-up time and discarded. The model output is available hourly."

- Section 2.3: could be separated into two: (i) surface observations / (ii) evaluation methodology

- We separated the section in two parts, as it seems clearer.

- L146: "Vigna di Valle is still" -> "Vigna de Valle is" (remove still)

- Done.

- L165: "environmental characteristics" seem to actually mean cloud   classification by the satellite method, is that correct? Please clarify text and use suitable terminology.

- We changed the sentence: "Table 3 shows, for each station and season, as well as for the whole year, the percentage of data in clear, contaminated and overcast conditions, classified by the satellite method of Section 2.1."

- L170: (language) "with the stations" -> "between different stations" ?

- Corrected.

- L174-175: somewhat unclear sentence, please clarify

- Rephrased: "The RAMS GHI forecast is available hourly, while the frequency of pyranometer observations and MSG-GHI estimate is every half an hour. Pyranometer observations and MSG-GHI estimates were considered hourly, at the same time of the RAMS forecast output."

- L178-179 + L188-189: why not use equation numbers?

- We added the equation numbers.

- Figure 3: I would suggest swapping the axes, so that pyranometer values are on the x-axis and estimated values on the y-axis. This makes values above the 1:1 line correspond to overestimation and vice versa, which is more logical. Also, I think it would be interesting to add this kind of figure for each station as a supplement or appendix as some readers will be interested in that information. Finally, the figure would be easier to read if grid lines and a legend were added, and if the point style would be modified so that points would not overlap (as much) in the busy areas of the plot.

- Figure 3 was changed according to this comment. The Figures for other stations will be added as a supplement to the paper (and are shown at the end of this answer for completeness).

- L196 / Fig. 3: clarify how the regression lines were determined

- Clarified. Linear regression is computed using the pyranometer values as $x$ and MSG-GHI estimation (Figure 3a) or RAMS-GHI one-day ahead hourly forecast (Figure 3b) as $y$. The black regression line is for clear sky, the red one is for cloudy conditions (both contaminated and overcast), the blue is for all the dataset. This has been clarified both into the text and adding a legend to Figure 3.

- L201-202 and L220: "it is apparent the larger scatter" -> (language) please rephrase (also similar sentence construction elsewhere)

- Rephrased: "The data for cloudy conditions of Figure 3a show larger deviations from their regression line compared to clear sky data." Also in line 220: "The RAMS-GHI forecast data show larger deviations from their regression line compared to MSG-GHI.". Also in lines 210-211 "b) the correlation coefficient for cloudy conditions is lower compared to clear sky data and shows….".

- L229-230: in point b, it needs to be emphasized that RAMS is a forecast and thus not directly comparable to MSG.

- Ok. We wrote: "For the latter point, however, it is emphasized that the MSG and RAMS performance cannot be directly compared because RAMS is a forecast, while MSG gives an estimate of the GHI from radiance observations".

- L256-257: It is unclear to the reader how the conclusion about clouds being the main source of error was made. Could this be elaborated?

- In the revised version of the paper there will be a reference to the supplement where we show the scatter plots of the GHI for the pyranometer and RAMS-GHI forecast. From these figures is apparent the over forecast of cloudy conditions by RAMS for the pyranometers of Paganella and Aosta (points in the lower part of the figures). We wrote: "The inspection of the model output for those stations reveals that the

main source of error was the over forecast of cloudy conditions, as shown by the scatter plots between the RAMS-GHI one-day hourly forecast and the corresponding pyranometer values for these stations, given as a supplement to this paper"

- L257-264: This seems to be mostly somewhat loose speculation, although things are expressed as hard facts. The evidence presented in the manuscript does not support all these statements. Therefore, I recommend rewriting, to use more careful statements.

-Lines 262-264 were removed, while the rest of the discussion was rewritten using more careful statements. We wrote: "It is not easy to find the reason for this behaviour, because several factors could be involved as errors in the physical and numerical parameterizations of the model, and errors in the initial and boundary conditions. Also, the 4 km horizontal resolution is not enough to resolve the fine orographic structures over the Alps (Aosta and Paganella) and over the Apennines (Monte Cimone), and their interaction with the atmosphere. "

- L268-274: On a general level, I believe the explanations presented here to be plausible. However, I also find that the authors focus too much on explanations that have to do with local orography and horizontal resolution. Could there be something else involved as well? For example, one factor that certainly plays a role here is the fact that mountain stations have more clouds and clouds are difficult for the satellite algorithm (as seen later on in Fig 6).

-We added the following reasons: "…b) The classification of sky conditions is more difficult where the soil is covered by snow and, because this condition is more frequent for mountainous stations, it increases the MSG-GHI error for those stations; c) The estimate of the hourly GHI by the MSG is more difficult in cloudy conditions (Figure 4), which are more frequent for mountainous stations."

- L287: RMSE -> rRMSE?

 - Corrected.

- L288-289: unclear to me what is meant by "statistic shows more clearly the impact of ..."

- Corrected. "… this analysis …."

- L294 + Fig 5 caption: "RAMS-GHI one-day forecast". Here (and elsewhere when mentioning RAMS one-day forecast) it would be important to emphasize that RMSE is based on hourly values of the day ahead forecast from RAMS. The present text leads me to think that values may be daily.

-   Clarified throughout the paper. We used "RAMS-GHI one-day hourly forecast" where appropriate.

- L301: I find it odd to say "This result is caused by RMSE statistics".

   - Corrected. "This result is caused by the large differences between the RAMS-GHI one-day hourly forecast and observations."

- L321: "all sky conditions, which showed" -> (suggest) "all sky conditions, which indirectly showed..."

- Changed.

- L329-331: Unclear how this explains the difference?

- We removed the sentence because it was misleading.

- L333: Not completely true, compare with L445-446

- Corrected. We considered only two classes: clear and cloudy.

- L344-346: Please clarify how the persistence forecast is created. Are values hour by hour assumed to be the same between the two days?

-Clarified. We wrote: "The one-day hourly persistence forecast was computed using hour by hour the observed values of the previous day".

- L345: I find the use of the short-version "1D" somewhat misleading (as it makes me think of one-dimensional) and therefore suggest writing it out: one-day.

-Ok. Corrected everywhere.

- Section 3.3: split into two separate sections? (i) Daily evaluation / (ii) MOS application

-We divided Section 3.3 in two sections according to this comment.

- L381-383: There may also be other sources of MBE

-We modified the sentence according to this comment. We wrote:

"The MOS technique improves the forecast/estimate of the GHI by reducing the MBE. The MBE is caused by several factors related to both modelling and observations. In the context of this paper the most important causes of MBE are: a) the approximations in the meteorological model and in the methodology used to estimate GHI from MSG data, and; b) the horizontal grid used to represent the real world, which smoothens the surface features causing systematic errors. Other contributions arise from small and undetected systematic errors in the observations, and from the not exact simultaneity of the three datasets (pyranometers, MSG-GHI, RAMS-GHI forecast). "

- L384: "The MOS consists of" -> (suggestion) "The MOS used here consists of"

-Ok. Corrected. The introduction on MOS has been moved in section 2.4.

- L389-393: Unclear how exactly this works, please clarify.

We clarified the methodology. We wrote "This method is a cross-validation method to assess how the MOS prediction will perform in practice. For each hour of a season, the dataset is divided in two parts: a) the actual data (or actual value), which is the value at the selected hour of the RAMS one-day hourly forecast (or the MSG hourly estimate of GHI) and the corresponding pyranometer observation, and: b) the training dataset, which is composed by all data in the season with the exception of the actual data. The Eqn. (5) is computed for the training dataset ($y$ is the pyranometer value and $x$ is the RAMS one-day hourly forecast or MSG hourly estimate of GHI), and it is applied to the actual data, which is the $x$, to give the corrected forecast ($y$). Because the MOS is computed starting from hourly data, the training period is all the season but one hour. This procedure was repeated for all the hourly data in the season, obtaining the time series of the corrected RAMS one-day hourly forecast and of the corrected MSG hourly estimation of the GHI. The RMSE and rRMSE were computed for the corrected forecast/estimate of the GHI. In this way, the data used for computing MOS is statistically independent from the dataset used for the verification."

- L402: Somewhat unclear how this conclusion can be made based on the above sentences.

-Ok. We removed the sentence because it is not a direct consequence of the above sentences.

- L436-440 / L441-446: the latter paragraph presents discussion on performance as function of cloud classification, while the previous contains similar information, but indirectly. Maybe the paragraphs could be combined into one, or the order be changed, to make more effective and convincing communication.

-We changed the order of the two paragraphs and we joined them. We wrote: "The cloud coverage has an important impact also on the RMSE of both MSG-GHI hourly estimate and RAMS-GHI one-day hourly forecast. The error is higher for cloudy conditions compared to clear sky. This is especially evident for RAMS because the RMSE averaged over all the stations varies from 91 W/m$^2$, to 191 W/m$^2$, and to 245 W/m$^2$ for clear, contaminated and overcast conditions, respectively; for MSG-GHI, the RMSE averaged over all stations varies from 68 W/m$^2$, to 123 W/m$^2$, and to 98 W/m$^2$ for clear, contaminated and overcast conditions, respectively. However, the analysis of the rRMSE reveals more clearly the impact of the cloud coverage on the performance. Both RAMS-GHI one-day hourly forecast and MSG-GHI hourly estimate show the largest rRMSE in winter and the lowest in summer, following the behaviour of the cloud coverage. "

**Supplemental material**

In the following, according to the suggestion of Reviewer #2, we show the scatterplots of the pyranometers and MSG-GHI hourly estimate (Figures 1-12 a) and the scatterplots of the pyranometers and RAMS-GHI one-day ahead hourly forecast (Figures 1-12 b) for all the stations considered in this paper. These Figures will be given as a supplement to the paper.

a)

[Figure]

b)

[revised manuscript text omitted]

---

## Author Response (AR2)

**Associate Editor Decision: Publish subject to minor revisions (Editor review)** (17 May 2017) by Stelios Kazadzis

Comments to the Author:
This is a very interesting work concerning the evaluation of
a. MSG/Seviri based GHI
b. RAMS based GHI
and c. RAMS-MOS (and persistence) based GHI

The work has been improved based on the reviewer's comments and the author's revision and it fits very well to the AMT journal.
I would suggest the publication of this paper to AMT after taking into account some minor comments below.

MSG-GHI inputs

The MSG relate section (2.1). This sections mainly describes the SICCS and Greuel, 2013 paper. In order to assess the use of this method to the selected Italian stations, some more information have to be provided about the input data used for each station.

For example Greuel, 2013 tables 1 and 2 describe the inputs used but then a choice of these for each station has to be made. e.g. if water or ice clouds were used, if the change of albedo was taken into account, also what water vapour value was used. Finally, if the station height has been taken into account in the radiative transfer modling calculations.

The SICCS algorithm is unaware of the stations, and no specific choices for these stations have been made. The cloud information (e.g. liquid or ice) comes from the CPP cloud property retrieval (i.e. from the satellite data), which is run prior to SICCS.
The information on surface albedo comes from a seasonally (8-day) varying MODIS-based climatology, interpolated to the MSG-SEVIRI grid. The information on total column water vapour comes from ERA-Interim. Surface elevation comes from the ETOPO2v2-2006 database. These inputs vary spatially and temporally (except elevation).
We propose to add the following text in the manuscript providing some more information on the inputs to the SICCS algorithm.

We wrote in Section 2.1 : "Other inputs include surface elevation from the ETOPO2v2-2006 database, monthly varying integrated atmospheric water vapour from the European Centre for Medium range Weather Forecast (ECMWF) ERA-Interim reanalysis, and 8-day varying surface albedo derived from Moderate-resolution Imaging Spectroradiometer (MODIS) data."

line 492 "more than halved" - you can provide the actual percentages
-The percentages have been added for the four seasons and for the whole year. As stated into the paper the percentage is more than halved if we exclude mountainous stations. Considering all stations, the RMSE of the MSG-GHI hourly estimate is "closer" to the RAMS-GHI one-day hourly forecast.
We wrote: "The RMSE of the MSG-GHI hourly estimate is more than halved compared to RAMS-GHI, with the exception of the mountainous stations where the RMSE of the two datasets are closer. In particular, excluding (including) the mountainous stations, the RMSE of the MSG-GHI hourly estimate is 49% (59%) of the RAMS-GHI one-day hourly forecast in winter; this percentage is 43% (49%) in spring, it is 54% (60%) in summer, it is 50% (57%) in fall, and it is 47% (52%) for the whole year."

Kosmopulos is actually Kosmopoulos
-Ok. Thank you and sorry for this mistake.

line 606 "big issues" I would rephrase that as despite the large discrepancies it is a challenging issue.
-We wrote: "Overall, the results of this paper show that the MSG-GHI estimate and the RAMS-GHI have large discrepancies with observations in cloudy conditions, and they are still challenging issues."

The cloud scenes are now divided to clear, contaminated and overcast based on MSG. They could be divided based on pyranometer data which represent more real conditions. Since, now it is difficult to be implemented, some discussion on this issue could be useful. For example that MSG failures on the cloud "typing" (clear, contaminated, overcast) will affect also the statistics.

for example contaminated data in theory are the most difficult ones to model as the hourly GHI is related with the percentage of time that the direct sun component is not attenuated by clouds. However largest RAMS deviations are shown in overcast data and maybe this has to do with the above cloud typing.
- Thank you for noting this point. We wrote in the Summary and Conclusion section: "It is important to note that the cloud scenes (clear, contaminated and overcast) are divided in this work based on MSG data. This classification could be done considering the pyranometers data, which are more representative of real conditions and this issue will be considered in future studies. Errors in the classification of sky conditions impact the results of this paper. For example, contaminated data are the most difficult ones to model as the hourly GHI is related with the percentage of time that the direct sun component is not attenuated by clouds. However, the results of this paper shows that the largest RAMS deviations are in overcast conditions and this could be caused, at least in part, by errors in cloud typing."

line 612-613 case to case, I think you mean location dependent.
-Corrected. "Considering also the variability of the RAMS performance from site to site, the usefulness of the RAMS forecast from an economic perspective is location dependent (Wittman et al. 2008)."